# CaCuTe: Casual Cubic-Model Technique for Faster Optimization

## Abstract

We establish a local $\mathcal{O}(k^{-2})$ rate for the gradient update $x^{k+1} = x^k - \nabla f(x^k)/\sqrt{H\|\nabla f(x^k)\|}$ under a $2H$-Hessian–Lipschitz assumption. Regime detection relies on Hessian–vector products, avoiding Hessian formation or factorization. Incorporating this certificate into cubic-regularized Newton (CRN) and an accelerated variant enables per-iterate switching between the cubic and gradient steps while preserving CRN's global guarantees. The technique achieves the lowest wall-clock time among compared baselines in our experiments. In the first-order setting, the technique yields a monotone, adaptive, parameter-free method that inherits the local $\mathcal{O}(k^{-2})$ rate. Despite backtracking, the method shows superior wall-clock performance. Additionally, we cover smoothness relaxations beyond classical gradient–Lipschitzness, enabling tighter bounds, including global $\mathcal{O}(k^{-2})$ rates. Finally, we generalize the technique to the stochastic setting.

## 1 Introduction

In this work, we are interested in solving the unconstrained minimization problem $\min_{x\in\mathbb{R}^d} f(x)$ where $f\colon \mathbb{R}^d \to \mathbb{R}$ is a twice-differentiable function with Lipschitz Hessian.

### 1.1 Motivation

The cubic regularized Newton (CRN) method (Griewank, 1981) naturally follows from the Lipschitz–Hessian assumption (Nesterov & Polyak, 2006). Its update can be written implicitly as

$$H\|x^{k+1} - x^k\|(x^{k+1} - x^k) = -\nabla f(x^k) - \nabla^2 f(x^k)(x^{k+1} - x^k)$$

where $H > 0$ is a constant, $\mathbf{I}$ is the identity matrix. When the quadratic term is negligible, i.e.

$$H\|x^{k+1} - x^k\|^3 \gg \langle \nabla^2 f(x^k)(x^{k+1} - x^k), x^{k+1} - x^k \rangle$$

an approximate update formula reads as

$$H\|x^{k+1} - x^k\|^3 \approx -\langle \nabla f(x^k), x^{k+1} - x^k \rangle \quad \text{or} \quad x^{k+1} \approx x^k - \frac{\nabla f(x^k)}{\sqrt{H\|\nabla f(x^k)\|}}$$

and raises a natural question:

*Can we provably avoid Hessian evaluation and inversion while retaining an $\mathcal{O}(k^{-2})$ rate?*

This affirmative result shows that the convergence is achievable with a first-order method and naturally leads to the next question.

*How to utilize Hessian smoothness for the first order methods?*

This question is central, since exact second-order steps are often infeasible in modern high-dimensional problems even when Hessian smoothness holds. Our approach leverages smoothness without forming or inverting Hessians. We also extend the framework to stochastic optimization, yielding practical algorithms for large-scale applications.

## 1.2 RELATED WORKS

**Cubic Regularization.** Beyond advances in Quasi-Newton methods (Scheinberg & Tang, 2016; Ghanbari & Scheinberg, 2018; Rodomanov & Nesterov, 2021a;b; Jin et al., 2022; Berahas et al., 2022; Kamzolov et al., 2023; Jin et al., 2024; Scieur, 2024; Wang et al., 2024), global convergence guarantees in the convex enhances by cubic regularization (Kamzolov et al., 2023; Scieur, 2024; Wang et al., 2024). However, those results have implicit (explicit for (Mishchenko, 2023)) update formula requiring additional line search in each iteration, involving matrix inversions. In contrast, our approach enables occasional skipping of these costly operations. It is worth mentioning (Doikov et al., 2023) enables only periodical Hessian computations. We also obtain a result similar to (Agafonov et al., 2025), which establishes global $\mathcal{O}(k^{-1})$ rate while having $\mathcal{O}(k^{-2})$ at the first iterates.

**Adaptive Gradient Methods.** Gradient descent is attractive for its simplicity and low per-iteration cost, the only hyperparameter is the step size: classical Lipschitz-based analyses pick a constant step to ensure monotone decrease, but this requires the (unknown) Lipschitz constant and is typically overly conservative in practice (Malitsky & Mishchenko, 2020). Backtracking line search (Nocedal & Wright, 2006), which gives theoretical convergence guarantee, is a parameter-free adaptive alternative to constant step sizes. However its try-increase nature can be avoided while keeping adaptivity without any extra computations under convexity and Lipschitz gradients (Malitsky & Mishchenko, 2020; 2024). Alternative adaptive methods with Polyak-stepsize (Polyak, 1987), requires a knowledge of a value of the objective's minimum.

**Smoothness.** Smoothness relaxations recently draw significant attention recently. One of such assumptions is $(L_0, L_1)$-*smoothness* originally introduced by (Zhang et al., 2020b) for twice differentiable functions, and extended to the class of differentiable but not necessarily twice differentiable functions (Zhang et al., 2020a; Chen et al., 2023). We introduce the assumption that the Hessian eigenvalue in the gradient direction grow linearly with the *square root of the optimality gap* instead *gradient norm*. Another closely related work (Mishkin et al., 2024) also study directional smoothness, but based on a quadratic model, in contrast to ours cubic one. Adaptive methods (Defazio & Mishchenko, 2023; Mishchenko & Defazio, 2024; Ivgi et al., 2023; Khaled et al., 2023) avoid smoothness assumption in their analysis in the convex setting as well as any line search.

**HVP Methods.** To the best of our knowledge Hessian vector products are used only for Hessian approximation (Jahani & Rusakov, 2022; Yao et al., 2021; 2018; Sen & Mohan, 2023) i.e. based on (Bekas et al., 2007). Except the closest work (Smee et al., 2025), which studies similar idea for non-convex deterministic setup. Also, a similar step size appears in (Thomsen & Doikov, 2024), which studies quadratic functions.

## 1.3 OUR CONTRIBUTION

1. **Cubic-Model Decrease.** In Section 3 we show that the gradient step ($x^{k+1} = x^k - \nabla f(x^k)/\sqrt{H_k \|\nabla f(x^k)\|}$ where $H_k$ is a local estimate of the Hessian–Lipschitz constant) yields a descent inequality that enables local $\mathcal{O}(k^{-2})$ rate. Under illustrative *vanishing directional curvature* condition (see (10)) it yields global $\mathcal{O}(k^{-2})$.

2. **Casual Cubic Newton (CaCuN).** In Section 4.1 we show that the descent inequality enables provable obviating the second order work (Hessian computation and factorization) for CRN (Nesterov & Polyak, 2006) type methods retaining global $\mathcal{O}(k^{-2})$ rate.

3. **Accelerated Casual Cubic Newton (AccCaCuN).** The same descent inequality extends to the accelerated setting in Section 4.2: AccCaCuN retains the global $\mathcal{O}(k^{-3})$ rate of accelerated CRN while provably obviating second-order work; under the illustrative *vanishing directional curvature* condition, no second-order work is needed for the global $\mathcal{O}(k^{-3})$ rate.

4. **Casual Cubic Adaptive Gradient Descent (CaCuAdGD).** A central contribution (Section 5) of our work addresses the infeasibility of directly applying second-order methods. We propose a step-size rule that uses a local estimate of the Hessian–Lipschitz constant and the Hessian eigenvalue in the gradient direction. Its computational cost is on the order of gradient evaluations. The rule yields a unified descent inequality that either (i) recovers the standard quadratic-model decrease or (ii) attains the cubic-model decrease, implying local $\mathcal{O}(k^{-2})$ and global $\mathcal{O}(k^{-1})$

rates. The method is monotone, parameter-free, and shows superior wall-clock performance in experiments compared to, e.g., AdGD (Malitsky & Mishchenko, 2020).

5. **Gradient-Directional Smoothness Relaxation.** In Section 5.3 we propose an assumption allowing the Hessian eigenvalue in the gradient direction to grow linearly with the square root of the optimality gap. It enables tighter convergence rates and, in particular, the global $\mathcal{O}(k^{-2})$ rate, e.g., for logistic regression under separability.

6. **Casual Cubic Stochastic Gradient Descent (CaCuSGD).** Under standard assumptions, having the mini-batch gradient and HVP estimators, our analysis (Section 6) recovers either the stochastic cubic–Newton decrease (Chayti et al., 2023) or the standard stochastic decrease under $L$-smoothness.

## 2  NOTATION AND MAIN ASSUMPTIONS

Our theory is based on the following assumption about second-order smoothness, which is also the key tool in proving the convergence of cubic Newton (Nesterov & Polyak, 2006).

**Assumption 2.1.** There exists a constant $H > 0$ such that for any $x, y \in \mathbb{R}^d$

$$f(y) \leq f(x) + \langle \nabla f(x), y - x \rangle + \frac{1}{2}\langle \nabla^2 f(x)(y - x), y - x \rangle + \frac{H}{3}\|y - x\|^3, \tag{1}$$

$$\|\nabla f(y) - \nabla f(x) - \nabla^2 f(x)(y - x)\| \leq H\|x - y\|^2. \tag{2}$$

Both inequalities hold if the Hessian of $f$ is $2H$-Lipschitz, i.e. $\|\nabla^2 f(x) - \nabla^2 f(y)\| \leq 2H\|x - y\|$ for all $x, y \in \mathbb{R}^d$.

The proof is given in (Nesterov & Polyak, 2006, Lemma 1).

**Assumption 2.2** (Convexity). Function $f : \mathbb{R}^d \to \mathbb{R}$ is convex, i.e. $\forall x, y \in \mathbb{R}^d$ it holds

$$f(y) \geq f(x) + \langle \nabla f(x), y - x \rangle. \tag{3}$$

**Assumption 2.3** (Bounded level set). Let $x^0 \in \mathbb{R}^d$ and $f : \mathbb{R}^d \to \mathbb{R}$ be continuous. The sublevel set

$$\mathcal{L}_0 := \{x \in \mathbb{R}^d : f(x) \leq f(x^0)\}$$

is bounded, i.e.,

$$D := \sup_{x \in \mathcal{L}_0} \|x - x^\star\| < \infty, \qquad x^\star \in \arg\min f.$$

The above assumptions are commonly used for analyzing the second order methods (Nesterov & Polyak, 2006; Nesterov, 2008; Doikov & Nesterov, 2020).

## 3  DESCENT ANALYSIS

In this section, we address contributions (1) and partially (4), i.e., the unified descent inequality and the corresponding rates. Since $f(x)$ has $2H$-Lipschitz–Hessian (Assumption 2.1), there exists a local value $H_k \leq H$ s.t.

$$f(x^+) \leq f(x^k) + \langle \nabla f(x^k), x^+ - x^k \rangle + \frac{1}{2}\langle x^+ - x^k, \nabla^2 f(x^k)(x^+ - x^k) \rangle + \frac{H_k}{3}\|x^+ - x^k\|^3.$$

By setting $x^+ = x^k - \frac{\nabla f(x^k)}{\sqrt{M_k\|\nabla f(x^k)\|}}$ for $M_k \geq H_k$ and $\alpha \in (0, 1)$ we then obtain

$$f\left(x^k - \frac{\nabla f(x^k)}{\sqrt{M_k\|\nabla f(x^k)\|}}\right) \leq f(x^k) - \frac{2(1 - \alpha)}{3\sqrt{M_k}}\|\nabla f(x^k)\|^{3/2}$$

$$+ \frac{1}{2M_k\|\nabla f(x^k)\|}\left[\langle \nabla f(x^k), \nabla^2 f(x^k)\nabla f(x^k)\rangle - \frac{4\alpha\sqrt{M_k}}{3}\|\nabla f(x^k)\|^{5/2}\right]. \tag{4}$$

The latter means that there exists

$$M_k = \max\left(H_k, \frac{9\langle\nabla f(x^k), \nabla^2 f(x^k)\nabla f(x^k)\rangle^2}{16\alpha^2\|\nabla f(x^k)\|^5}\right), \quad \text{s.t.} \tag{5}$$

$$f\left(x^k - \frac{\nabla f(x^k)}{\sqrt{M_k\|\nabla f(x^k)\|}}\right) \leq f(x^k) - \frac{2(1-\alpha)}{3\sqrt{M_k}}\|\nabla f(x^k)\|^{3/2}. \tag{6}$$

Rule (5) forms the step $x^{k+1} = x^k - \frac{\nabla f(x^k)}{\sqrt{M_k\|\nabla f(x^k)\|}}$. It uses (a) a Hessian–gradient product, computable at (asymptotically) the cost of a gradient without forming the Hessian, and (b) the directional Hessian–Lipschitz constant $H_k$, obtained via backtracking whose per-iteration cost is also on the order of one gradient evaluation.

Inequality (6), together with (5), yields the *unified descent inequality*. It specializes for the two cases implied by (5) as follows:

*Case I (cubic-model decrease):* If $M_k = H_k$, then (6) becomes

$$f(x^{k+1}) \leq f(x^k) - \frac{2(1-\alpha)}{3\sqrt{H_k}}\|\nabla f(x^k)\|^{3/2}, \tag{7}$$

which yields the usual cubic-model decrease which corresponds to the $\mathcal{O}(k^{-2})$ regime by Lemma A.4.

*Case II (quadratic-model decrease):* If $M_k = \frac{9\langle\nabla f(x^k), \nabla^2 f(x^k)\nabla f(x^k)\rangle^2}{16\alpha^2\|\nabla f(x^k)\|^5}$ then (6) becomes

$$f(x^{k+1}) \leq f(x^k) - \frac{8(1-\alpha)\alpha}{9L_k}\|\nabla f(x^k)\|^2, \tag{8}$$

where $L_k = \frac{\langle\nabla f(x^k), \nabla^2 f(x^k)\nabla f(x^k)\rangle}{\|\nabla f(x^k)\|^2}$, it yields to the classical quadratic-model decrease which corresponds to the $\mathcal{O}(k^{-1})$ regime by Lemma A.5.

Note that the bound $L_k \leq L$ required in Lemma A.5 holds under our standing assumptions. Because $\nabla^2 f$ is $2H$-Lipschitz and, by monotonicity (6) together with Assumption 2.3, the iterates remain in the compact sublevel set $\mathcal{L}_0$, we have

$$L := \sup_{x\in\mathcal{L}_0}\|\nabla^2 f(x)\| < \infty,$$

and hence $L_k \leq L$ for all $k$. In particular, $f$ is $L$-smooth along the trajectory of a monotone method.

## 4 REGULARIZED NEWTON METHODS

In this section, we show that the CRN step can be replaced by a gradient step when *Case I: cubic-model decrease* holds.

Section 3 establishes that there *exists* some $M_k \geq H_k$ for which (6) holds, although $M_k$ is not known explicitly. When a global $H$ from Assumption 2.1 is available, we simply *test* (6) with $M_k = H$. This provides a *sufficient* condition only: if (6) holds with $M_k = H$, then the gradient step attains the cubic-model decrease at $x^k$; otherwise, the cubic-model decrease is obtained by the CRN step. Thus correctness is preserved (we never accept an invalid first-order step), and we skip second-order work whenever this sufficient test passes.

We will specify $\alpha$ in (6) below so that the resulting bounds coincide (up to constants) with those for CRN (Nesterov & Polyak, 2006) and accelerated CRN (Nesterov, 2008).

### 4.1 CASUAL CUBIC NEWTON

In order to align our bound up to constants with the standard cubic Newton convergence rate one checks inequality (6) with $\alpha = 1 - \frac{1}{\sqrt{3}}$ and $M_k = H$. If it holds then $x^{k+1} = x^k - \frac{\nabla f(x^k)}{\sqrt{H\|\nabla f(x^k)\|}}$ implies CRN-like decrease

$$f(x^{k+1}) \leq f(x^k) - \frac{2(1-\alpha)}{3\sqrt{M_k}}\|\nabla f(x^k)\|^{3/2} = f(x^k) - \frac{1}{\sqrt{2H}}\left(\frac{2}{3}\right)^{3/2}\|\nabla f(x^k)\|^{3/2} \tag{9}$$

and pure CRN step is performed otherwise (Algorithm 1).

---

**Algorithm 1** Casual Cubic Newton (CaCuN)

---

1: **Input:** $x^0 \in \mathbb{R}^d$, $H > 0$
2: **for** $k = 1, 2, \dots$ **do**
3:     **if** $f\left(x^k - \frac{\nabla f(x^k)}{\sqrt{H\|\nabla f(x^k)\|}}\right) \leq f(x^k) - \frac{1}{\sqrt{2H}}\left(\frac{2}{3}\right)^{3/2}\|\nabla f(x^k)\|^{3/2}$ **then**
4:         $x^{k+1} = x^k - \frac{\nabla f(x^k)}{\sqrt{H\|\nabla f(x^k)\|}}$
5:     **else**
6:         $x^{k+1} = x^k - (\nabla^2 f(x^k) + H\|x^{k+1} - x^k\|\mathbf{I})^{-1}\nabla f(x^k),$         ▷ CRN step

---

**Algorithm 2** Accelerated Casual Cubic Newton (AccCaCuN)

---

1: **Input:** $y^0 = x^0 \in \mathbb{R}^d$, $H > 0$
2: **Init:**
3: **if** $f\left(y^0 - \frac{\nabla f(y^0)}{\sqrt{H\|\nabla f(y^0)\|}}\right) \leq f(y^0) - \frac{\sqrt{2}}{3\sqrt{H}}\|\nabla f(y^0)\|^{3/2}$ **then**
4:     $x^1 = y^0 - \frac{\nabla f(y^0)}{\sqrt{H\|\nabla f(y^0)\|}}$
5: **else**
6:     $x^1 = x^0 - (\nabla^2 f(x^0) + 2H\|x^1 - x^0\|\mathbf{I})^{-1}\nabla f(x^0)$
7: $\psi_1(x) = \frac{N}{3}\|x - x_0\|^3 + f(x^1)$
8: **for** $k = 1, 2, \dots$ **do**
9:     $v^k = \arg\min_{x \in \mathbb{R}^d} \psi_k(x)$
10:     $y^k = \frac{k}{k+3}x^k + \frac{3}{k+3}v^k$
11:     **if** $f\left(y^k - \frac{\nabla f(y^k)}{\sqrt{H\|\nabla f(y^k)\|}}\right) \leq f(y^k) - \frac{1}{\sqrt{3H}}\|\nabla f(y^k)\|^{3/2}$ **then**
12:         $x^{k+1} = y^k - \frac{\nabla f(y^k)}{\sqrt{H\|\nabla f(y^k)\|}}$
13:         $\psi_{k+1}(x) = \psi_k(x) + \frac{(k+1)(k+2)}{2}\left[f(y^k) + \langle \nabla f(y^k), x - y^k \rangle\right]$
14:     **else**
15:         $x^{k+1} = x^k - (\nabla^2 f(x^k) + 2H\|x^{k+1} - x^k\|\mathbf{I})^{-1}\nabla f(x^k)$     ▷ CRN step
16:         $\psi_{k+1}(x) = \psi_k(x) + \frac{(k+1)(k+2)}{2}\left[f(x^{k+1}) + \langle \nabla f(x^{k+1}), x - x^{k+1} \rangle\right]$

---

*Remark* 4.1. One may test any value $M_k \leq H$. Using a smaller $M_k$ makes (6) stricter, so the gradient step is accepted less often (but never incorrectly); if the test fails, we fall back to the CRN step.

**Theorem 4.2** ($\downarrow$)**.** *Let Assumptions 2.1 to 2.3 hold. Then the iterates of Algorithm 1 satisfy the following*

    *1. If $f(x^0) - f_\star \geq 3HD^3$, then*

$$f(x^1) - f_\star \leq 3HD^3.$$

    *2. If $f(x^0) - f_\star \leq 3HD^3$, then for all $k \geq 0$,*

$$f(x^k) - f_\star \leq \frac{3HD^3}{(1 + \frac{k}{3})^2}.$$

The method retains the global rate of CRN while *provably skipping second-order work when possible*.

### 4.2 ACCELERATED CASUAL CUBIC NEWTON

Similarly, we match the constants of accelerated cubic Newton from Nesterov (2008) by checking (6) with $\alpha = 1 - \frac{1}{\sqrt{2}}$ at the first iteration and $\alpha = \frac{1}{2}$ thereafter. The resulting rates are the same up to constants, but whenever this sufficient condition holds, we replace the cubic step with a gradient step and thus skip second-order work on those iterations.

**Theorem 4.3** ($\downarrow$). *Let Assumptions 2.1 to 2.3 hold. Then the iterates of Algorithm 2 satisfy the following*

$$f(x^k) - f(x^*) \leq \frac{14 \cdot 2H \|x^0 - x^*\|^3}{k(k+1)(k+2)}.$$

Same technique is also applicable for regularized global Newton and AdaN (Mishchenko, 2023). Analogously to its adaptive modification AdaN+ (Mishchenko, 2023), we propose a heuristic extension that uses gradient-based backtracking to calibrate the local Hessian–Lipschitz constant. Results are in Section D.

## 5 FIRST-ORDER METHOD

In this section we consider first-order update $x^{k+1} = x^k - \nabla f(x^k)/\sqrt{M_k \|\nabla f(x^k)\|}$. We additionally make assumptions on gradient smoothness guaranteeing decrease when cubic regime is not feasible.

### 5.1 VANISHING DIRECTIONAL CURVATURE

We revisit (4) with $\alpha = \frac{3}{4}$ and consider the following *illustrative* growth condition on the curvature in the gradient direction:

$$\frac{\langle \nabla f(x^k), \nabla^2 f(x^k) \nabla f(x^k) \rangle}{\|\nabla f(x^k)\|^2} \leq \sqrt{C \|\nabla f(x^k)\|} \qquad \text{for some } C > 0. \tag{10}$$

Let $M \geq \max\{H, C\}$ and take the gradient step

$$x^{k+1} = x^k - \frac{\nabla f(x^k)}{\sqrt{M \|\nabla f(x^k)\|}}.$$

Plugging (10) into (4) yields the cubic-model decrease

$$f(x^{k+1}) \leq f(x^k) - \frac{1}{6\sqrt{M}} \|\nabla f(x^k)\|^{3/2}. \tag{11}$$

Next, Lemma A.4 assumes a decrease of the form

$$f(x^{k+1}) \leq f(x^k) - \frac{1}{3\sqrt{H_k}} \|\nabla f(x^k)\|^{3/2} \quad \text{with } H_k \leq H \text{ for all } k,$$

so the constant

$$H_k \equiv 4M$$

so that $\frac{1}{3\sqrt{H_k}} = \frac{1}{6\sqrt{M}}$ matches the lemma's premise due to (11). Therefore, under Assumptions 2.2 and 2.3, Lemma A.4 gives a global $\mathcal{O}(k^{-2})$ rate.

While the class in (10) is uncommon in modern practice, it is nonempty and includes several simple families with explicit constants and Lipschitz Hessians. We provide examples below.

**Example 5.1** ($\downarrow$). *Let $f(x) = \frac{H}{3} \|x\|^3$, $H > 0$. Then $\nabla^2 f$ is $2H$-Lipschitz and (10) holds with $C = 4H$.*

**Example 5.2** ($\downarrow$). *Let $f(x) = \sum_{i=1}^d \frac{H_i |x_i|^3}{3}$, $H_i > 0$. Then $\nabla^2 f$ is $2H$-Lipschitz and (10) holds with $C = 4H$, where $H = \max_i H_i$.*

**Example 5.3** (Logistic function $\downarrow$). *Let $a \in \mathbb{R}^d \setminus \{0\}$ and $f(x) = \log(1 + e^{-a^\top x})$. Then $\nabla^2 f$ is $\frac{\|a\|^3}{6\sqrt{3}}$-Lipschitz and (10) holds with $C = \frac{4}{27} \|a\|^3$.*

### 5.2 CASUAL CUBIC ADAPTIVE GRADIENT DESCENT (CACUADGD)

In addition to the uncommon class in (10), we consider the standard $L$-smooth setting.

*Remark* 5.4. Since $\nabla^2 f$ is $2H$-Lipschitz and, by monotonicity with Assumption 2.3, the iterates remain in the compact sublevel set $\mathcal{L}_0$, we have $L := \sup_{x \in \mathcal{L}_0} \|\nabla^2 f(x)\| < \infty$, so $\nabla f$ is $L$-Lipschitz on $\mathcal{L}_0$.

Even though a global (worst-case) $L$ exists, it can be larger than the *directional* curvature scale. In particular, when there exists some bounded $C$ such that $L_k \leq \sqrt{C\|\nabla f(x^k)\|} \ll L$, then the directional curvature bound (cf. (10)) becomes more informative than the global $L$ bound, and the resulting decrease yields the improved rate via Lemma A.4.

This motivates two regimes to analyze jointly: (i) a *growth* regime, where directional curvature scales with the iterate's difficulty, and (ii) a *bounded* regime, where directional curvature is controlled by a constant. We unify these regimes via a single *directional growth* Assumption 5.11 that relaxes standard $L$-smoothness and, as a special case, recovers global $\mathcal{O}(k^{-2})$. Building on this, we introduce CaCuAdGD (Algorithm 3), a monotone, parameter-free adaptive scheme.

**Idea.** At each iteration, CaCuAdGD estimates the local Hessian smoothness *along the gradient direction* via a backtracking procedure that relies only on Hessian–vector products (no explicit Hessian computations). This yields an *automatic switch* between: (i) a *bounded-Hessian* regime, where the quadratic model driven decrease holds; and (ii) a *Hessian-Lipschitz* (cubic) regime, where a cubic-model controls the remainder.

---

**Algorithm 3** Casual Cubic Adaptive Gradient Descent (CaCuAdGD)

---

1: **Input:** $x^0 \in \mathbb{R}^d$, $H > 0$, $\alpha \in (0,1)$
2: **for** $k = 0, 1, \dots$ **do**
3:     $\widehat{H} = \frac{9\langle \nabla f(x^k), \nabla^2 f(x^k)\nabla f(x^k)\rangle^2}{16\alpha^2 \|\nabla f(x^k)\|^5}$, $H = H/16$
4:     **while** $f\left(x^k - \frac{\nabla f(x^k)}{\sqrt{H\|\nabla f(x^k)\|}}\right) \geq f(x^k) + \frac{1}{2}\frac{\langle \nabla f(x^k), \nabla^2 f(x^k)\nabla f(x^k)\rangle}{H\|\nabla f(x^k)\|} - \frac{2\|\nabla f(x^k)\|^{3/2}}{3\sqrt{H}}$ **do**
5:       $H = 2H$
6:     $x^{k+1} = x^k - \frac{\nabla f(x^k)}{\sqrt{\max\{H, \widehat{H}\}\|\nabla f(x^k)\|}}$

---

### 5.2.1 LIPSCHITZ GRADIENT

In this section we consider the *bounded* regime, where directional curvature is controlled by a constant. It is formalized by the following assumption.

**Assumption 5.5** (*L*-smoothness). Function $f : \mathbb{R}^d \to \mathbb{R}$ is $L$-smooth, i.e., for all $x \in \mathbb{R}^d$ we have

$$\|\nabla^2 f(x)\| \leq L. \tag{12}$$

Setting $\alpha = 0.5$ (5) yields $M_k = \max\left(H_k, \frac{9\langle \nabla f(x^k), \nabla^2 f(x^k)\nabla f(x^k)\rangle^2}{4\|\nabla f(x^k)\|^5}\right)$ and either

$$f(x^{k+1}) \leq f(x^k) - \frac{1}{3\sqrt{H_k}}\|\nabla f(x^k)\|^{3/2} \quad \text{or} \quad f(x^{k+1}) \leq f(x^k) - \frac{2}{9L_k}\|\nabla f(x^k)\|^2,$$

where $L_k = \frac{\langle \nabla f(x^k), \nabla^2 f(x^k)\nabla f(x^k)\rangle}{\|\nabla f(x^k)\|^2} \leq L$.

Since the optimality gap $f(x^k) - f_\star$ is monotonically decreases, iterates split into 2 sets.

**Lemma 5.6** ($\downarrow$). *Let Assumptions 2.1 to 2.3 and 5.5 hold. Then the iterates generated by Algorithm 3 with $\alpha = 0.5$ satisfy the following*

$$\sqrt{f(x^k) - f_\star} \geq \frac{3}{2}\frac{LD^2}{\sqrt{HD^3}}, \quad \text{then} \quad f(x^{k+1}) - f_\star \leq f(x^k) - f_\star - \frac{1}{3\sqrt{HD^3}}\left(f(x^k) - f_\star\right)^{1.5} \tag{13}$$

$$\sqrt{f(x^k) - f_\star} \leq \frac{3}{2}\frac{LD^2}{\sqrt{HD^3}}, \quad \text{then} \quad f(x^{k+1}) - f_\star \leq f(x^k) - f_\star - \frac{2}{9LD^2}\left(f(x^k) - f_\star\right)^2. \tag{14}$$

Inequality (13) immediately implies

**Corollary 5.7** ($\downarrow$). *Let Assumptions 2.1 to 2.3 and 5.5 hold. Then the iterates generated by Algorithm 3 with $\alpha = 0.5$ satisfy for all $k$ s.t. $\left(f(x^k) - f_\star\right)^{1/2} \geq \frac{3}{2}\frac{LD^2}{\sqrt{HD^3}}$ the following*

$$f(x^{k+1}) - f_\star \leq \frac{\left(9\sqrt{HD^3} + 3\sqrt{f(x^0) - f_\star}\right)^2}{k^2}.$$

If the desired accuracy is not reached in the accelerated regime, the global rate is as follows.

**Theorem 5.8** ($\downarrow$). *Let Assumptions 2.1 to 2.3 and 5.5 hold. Then the iterates generated by Algorithm 3 with $\alpha = 0.5$ satisfy for all $k$ s.t. $\left(f(x^k) - f_\star\right)^{1/2} \leq \frac{3}{2}\frac{LD^2}{\sqrt{HD^3}}$ the following*

$$f(x^k) - f_\star \leq \frac{27}{2}\frac{LD^2}{k + LD^2 \cdot \Phi(f(x^0) - f_\star)},$$

*where $\Phi(\xi) := \frac{9}{2}\frac{1}{\xi} + \frac{6\sqrt{H}}{L\sqrt{D}}\frac{1}{\sqrt{\xi}}$.*

*Remark* 5.9. In practice, the two-regime split may not hold, meaning, that the algorithm performs the best possible step enabling the cubic-model decrease even for later iterates.

**Corollary 5.10.** *Let Assumptions 2.1 to 2.3 and 5.5 hold. Then the total number of iterations needed to reach $f(x^K) - f_\star \leq \varepsilon$ satisfies*

$$K = \mathcal{O}\left(\frac{LD^2}{\varepsilon} + \frac{(\sqrt{HD^3} + \sqrt{f(x^0) - f_\star})^2}{\sqrt{\varepsilon}}\right).$$

### 5.3 RELAXED SMOOTHNESS

Similarly to $(L_0, L_1)$-*smoothness* originally introduced by (Zhang et al., 2020b) for twice differentiable functions, we allow the norm of the Hessian of the objective to increase linearly with the growth of square root of the optimality gap. We show that this assumption holds, for example, in logistic regression under separability. Importantly, it controls *only* the directional growth of the Hessian.

**Assumption 5.11** (Directional $L_0, L_1$-smoothness). Function $f : \mathbb{R}^d \to \mathbb{R}$ is $L$-smooth, i.e., for all $x \in \mathbb{R}^d$ we have

$$\frac{\langle \nabla f(x), \nabla^2 f(x)\nabla f(x)\rangle}{\|\nabla f(x)\|^2} \leq L_0 + L_1(f(x) - f_\star)^{1/2}. \tag{15}$$

*Remark* 5.12. Assumption 5.11 is a relaxed version of the global $\|\nabla^2 f(x)\| \leq L_0 + L_1\left(f(x) - f_\star\right)^{1/2}$.

**Example 5.13** (Logistic function, $L_0 = 0$ $\downarrow$). Let $a \in \mathbb{R}^d \setminus \{0\}$, $s = \langle a, x\rangle$, and $f(x) = \log(1 + e^{-s})$. Then $f_\star = 0$ and for all $x \in \mathbb{R}^d$,

$$\frac{\langle \nabla f(x), \nabla^2 f(x)\nabla f(x)\rangle}{\|\nabla f(x)\|^2} \leq \frac{2}{3\sqrt{3}}\|a\|^2\left(f(x) - f_\star\right)^{1/2}.$$

**Example 5.14** (Empirical logistic, $L_0 = 0$ under separability $\downarrow$). Let there exists $v \in \mathbb{R}^d$ and $\gamma > 0$ s.t. $y_i\langle a_i, v\rangle \geq \gamma$ $\forall i$, then for $f(x) = \frac{1}{n}\sum_{i=1}^n \log\left(1 + e^{-y_i\langle a_i, x\rangle}\right)$ one has $f_\star = 0$ and $\forall x \in \mathbb{R}^d$,

$$\frac{\langle \nabla f(x), \nabla^2 f(x)\nabla f(x)\rangle}{\|\nabla f(x)\|^2} \leq \frac{2}{3\sqrt{3}}\left(\frac{1}{n}\sum_{i=1}^n \|a_i\|^4\right)^{1/2}\left(f(x) - f_\star\right)^{1/2}.$$

The proof of the following result mainly follows the arguments above, as the iterates split into cases where either vanishing curvature or standard smoothness holds.

**Corollary 5.15** ($\downarrow$). *Let Assumptions 2.2, 2.3, 5.5 and 5.11 hold. Then the total number of iterations needed to reach $f(x^K) - f_\star \leq \varepsilon$ satisfies*

$$K = \mathcal{O}\left(\frac{L_0 D^2}{\varepsilon} + \frac{\left(L_1 D^2 + \sqrt{HD^3} + \sqrt{f_0 - f_*}\right)^2}{\sqrt{\varepsilon}}\right).$$

When the method is *far* from the solution, the $L_1$ term in Assumption 5.11 can dominate and drive faster decrease; when it is *close* to the solution (so $f(x^k as) - f_\star$ is small), Assumption 5.11 effectively reduces to a local $L_0$-smoothness assumption (*not* with the global worst-case $L$). $L_0$ is the Lipschitz constant on the smaller sublevel set reached by the algorithm, and can be much smaller than the global bound. If $L_0 = 0$ in Assumption 5.11 the rate specializes to global $\mathcal{O}(k^{-2})$.

### 5.4 COMPARISON WITH AdGD

CaCuAdGD uses gradient steps $x^{k+1} = x^k - \lambda_k \nabla f(x^k)$ with

$$\lambda_k = \min\left\{\frac{16\alpha^2}{9L_k}, \frac{1}{\sqrt{H_k\|\nabla f(x^k)\|}}\right\},$$

while AdGD employs

$$\lambda_k = \min\left\{\frac{1}{2L_{k-1}}, \sqrt{1+\theta_{k-1}}\lambda_{k-1}\right\}, \qquad \theta_{k-1} := \frac{\lambda_{k-1}}{\lambda_{k-2}}.$$

**Step size growth.** When the local smoothness improves sharply (i.e., $L_k \ll L_{k-1}$), AdGD can increase $\lambda_k$ only multiplicatively via $\sqrt{1+\theta_{k-1}}$, which may take several iterations to reach the new scale $L_k$. In contrast, CaCuAdGD adapts in a single step $H_k\|\nabla f(x^k)\|$ vanishes near a solution, so the step is governed by $L_k$ (fast expansion to the local scale). Far from the solution, when $H_k\|\nabla f(x^k)\|$ is large, the cubic-model decrease (7) is guaranteed which corresponds to a local $O(1/k^2)$ decay.

**Choice of $\alpha$.** The parameter $\alpha \in (0,1)$ control the trade-off between the two regimes. Smaller $\alpha$ tightens $\frac{16\alpha^2}{9L_k}$ and triggers the quadratic regime earlier (more conservative). Whereas larger $\alpha$ loosens it, making the steps larger and lets the cubic regime act more often. At the same time, the decrease in the quadratic regime scales with $\alpha(1-\alpha)$ (8), so increasing $\alpha$ beyond $1/2$ reduces this per-step guarantee. Because the time spent in each regime is problem dependent, selecting $\alpha$ a priori is nontrivial. Our analysis allows any $\alpha \in (0,1)$; in practice, values slightly below $3/4$ (i.e. 0.7) perform consistently well on various problems, making the method effectively parameter free. Also, in the quadratic regime, the allowed step (the constant multiplying $1/L_k$) is larger for CaCuAdGD ($\approx 0.87$), compared with AdProxGD (Malitsky & Mishchenko, 2024) ($1/\sqrt{2}$) and AdGD (0.5).

## 6 STOCHASTIC EXTENSION

In this section we consider

$$\min_{x \in \mathbb{R}^d} f(x) = \mathbb{E}_\xi f_\xi(x),$$

particularly, a finite sum minimization $f(x) = \frac{1}{n}\sum_{i=1}^n f_i(x)$ where each $f_i : \mathbb{R}^d \to \mathbb{R}$ is a per-sample loss. Here, $\xi$ denotes a (multi)set of sampled indices $S \subseteq [n]$ (a mini-batch), so $f_\xi(x) \equiv f_S(x) = \frac{1}{|S|}\sum_{i \in S} f_i(x)$.

**Assumption 6.1.** Stochastic gradient $g(x,\xi)$ is an unbiased estimator of $\nabla f(x)$ with bounded variance, i.e., $\mathbb{E}_\xi[g(x,\xi)] = \nabla f(x)$ and for $\sigma_g \geq 0$

$$\mathbb{E}_\xi \|g(x,\xi) - \nabla f(x)\|^2 \leq \sigma_g^2, \tag{16}$$

stochastic Hessian $H(x,\xi)$ has bounded third moment

$$\mathbb{E}_\xi \|H(x,\xi) - \nabla^2 f(x)\|^3 \leq \sigma_H^3. \tag{17}$$

Moreover, $f_\xi$ is $L$-smooth, i.e., $\forall x \in \mathbb{R}^d$

$$\|H(x,\xi)\| \leq L. \tag{18}$$

Also, $f$ and $f_\xi$ $2H$-smooth Hessian , i.e., $\forall x,y \in \mathbb{R}^d$

$$f_{(\xi)}(x) \leq f_{(\xi)}(y) + \langle \nabla f_{(\xi)}(y), x-y \rangle + \tfrac{1}{2}\langle \nabla^2 f_{(\xi)}(y)(x-y), x-y \rangle + \tfrac{H}{3}\|x-y\|^3. \tag{19}$$

These assumptions are standard (Gorbunov et al., 2020; Chayti et al., 2023).

*Remark* 6.2. In the algorithm we *do not* form $H(x,\xi)$ explicitly; we only require stochastic Hessian–gradient products $H(x,\xi)g(x,\xi)$. Since, to the best of our knowledge, there is no standard assumption tailored exactly to this product, we use the standard third-moment bound above. However, our analysis also suffices under the following non-standard second-moment bound:

$$\mathbb{E}_\xi\left[\|(\nabla^2 f(x) - H(x,\xi))g(x,\xi)\|^2\right] \leq \sigma_P^2.$$

For brevity we write $g^k := g(x^k, \xi_k)$ and $H^k := H(x^k, \xi_k)$. The stochastic extension is below.

**Theorem 6.3** ($\downarrow$). *Let Assumption 6.1 holds. Assume that $\widehat{L} \geq 3L$ and $\widehat{H} \geq H$. Then the method (CaCuSGD)*

$$x^{k+1} = x^k - g^k/\sqrt{M_k\|g^k\|}, \quad M_k := \begin{cases} \widehat{L}^2\|g^k\|^{-1}, & if \quad 4\langle g^k, H^k g^k\rangle^2\|g^k\|^{-5} \geq \widehat{H}, \\ \widehat{H}, & otherwise, \end{cases} \quad satisfies$$

$$\mathbb{E}f(x^{k+1}) \leq \mathbb{E}f(x^k) - \frac{1}{4\widehat{L}}\mathbb{E}\|\nabla f(x^k)\|^2 + \left(\frac{2L}{\widehat{L}^2} + \frac{2L^2}{3\widehat{L}^3}\right)\sigma_g^2 + \frac{\sigma_H^3}{6\widehat{H}^2}, \qquad if \frac{4\langle g^k, H^k g^k\rangle^2}{\|g^k\|^{-5}} \geq \widehat{H},$$

$$\mathbb{E}f(x^{k+1}) \leq \mathbb{E}f(x^k) - \frac{1}{3\sqrt{\widehat{H}}}\mathbb{E}\|\nabla f(x^k)\|^{3/2} + \frac{7}{4\sqrt{\widehat{H}}}\sigma_g^{3/2} + \frac{32}{3\widehat{H}^2}\sigma_H^3, \qquad otherwise.$$

Telescoping each these one-step bounds solely yields one of the standard results. Because iterates can switch regimes and stochastic errors mix, we state per-iteration bounds.

The two regimes mirror the stochastic decrease results for standard SGD (Gorbunov et al., 2020) and stochastic CRN (Chayti et al., 2023). The cubic phase provide irreducible but lower-power variance floor $\sigma_g^{3/2}$, while the quadratic phase reproduces the SGD behavior: increasing $\widehat{L}$ (i.e., using a smaller stepsize) reduces the variance-driven term, enabling convergence to arbitrary accuracy. In practice, the cubic regime shows a faster transient toward the noise floor.

# 7 SELECTED EXPERIMENTS

We evaluate on logistic regression (covtype, w8a, mushrooms) and a synthetic log-sum-exp (LSE) problem; setup, hyperparameters, and extended results are in Section G. Non-adaptive CaCuN/AccCaCuN match CRN(Cubic)/Accelerated CRN(AccCubic) in iterations while reducing wall-clock. CaCuAdGD outperforms AdGD on hard tasks and remains competitive with Polyak and NesterovLS. Adaptive CaCuAdaN+ is strong on logistic but unstable on LSE since $H$ is under-estimated. In the stochastic setting, adaptive CaCuAdaN is not worse than its origin. CaCuSGD reaches the noise floor faster than SGD; even with conservative steps (large $\widehat{L}$), CaCuSGD attains the same or smaller floor sooner (effectively acting as an implicit scheduler). Extended experimental results are in Section G. Small $\ell_2 = 1e - 7$ regularization is used in Figures 4 and 5.

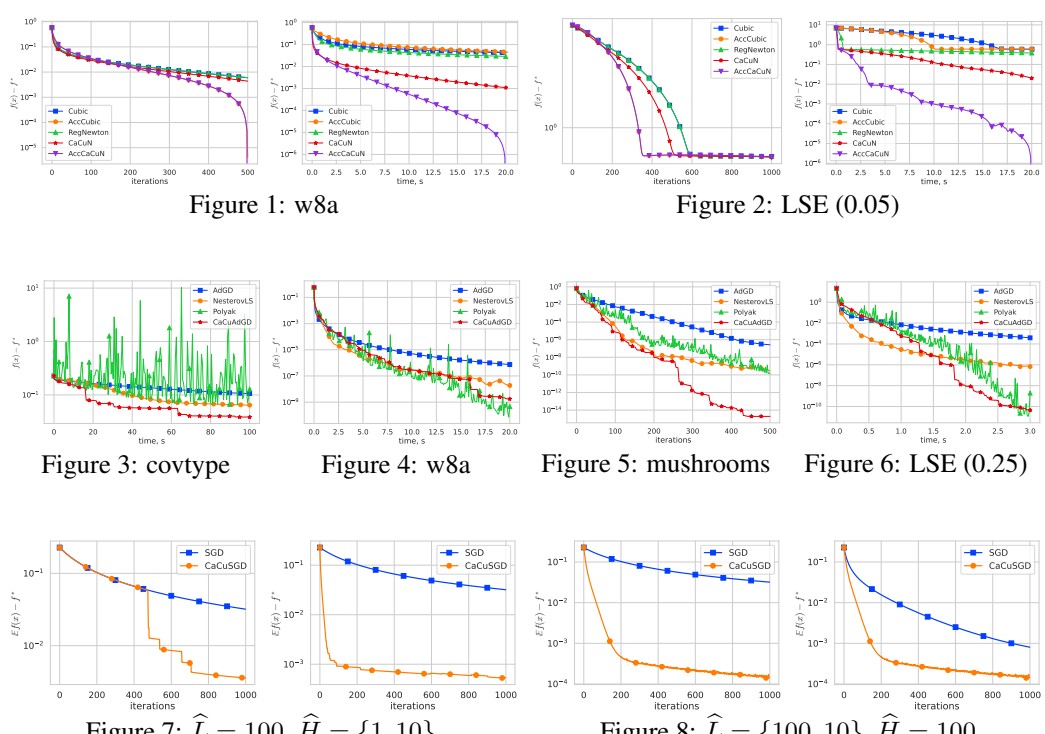

Figure 1: w8a

Figure 2: LSE (0.05)

Figure 3: covtype

Figure 4: w8a

Figure 5: mushrooms

Figure 6: LSE (0.25)

Figure 7: $\widehat{L} = 100$, $\widehat{H} = \{1, 10\}$

Figure 8: $\widehat{L} = \{100, 10\}$, $\widehat{H} = 100$

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

## A  AUXILIARY RESULTS

**Lemma A.1** (Young's inequality). *Let $p, q > 1$ with $\frac{1}{p} + \frac{1}{q} = 1$. For all $a, b \geq 0$,*

$$ab \leq \frac{a^p}{p} + \frac{b^q}{q}.$$

**Lemma A.2.** *For all $b > 0, \xi > 0, \beta > 0$,*

$$\frac{b}{\sqrt{\xi}} \leq \frac{\beta}{\xi} + \frac{b^2}{4\beta}.$$

*Proof.* Young's inequality $2xy \leq x^2 + y^2$ with $x = \sqrt{\beta/\xi}$ and $y = b/(2\sqrt{\beta})$ gives $\frac{b}{\sqrt{\xi}} = 2xy \leq x^2 + y^2 = \frac{\beta}{\xi} + \frac{b^2}{4\beta}$. □

**Lemma A.3.** *For all $a, b \in \mathbb{R}^d$ and any norm $\|\cdot\|$,*

$$\|a\|^{3/2} \leq \sqrt{2}\big(\|a - b\|^{3/2} + \|b\|^{3/2}\big).$$

*Equivalently,*

$$-\|a - b\|^{3/2} \leq -\tfrac{1}{\sqrt{2}}\|a\|^{3/2} + \|b\|^{3/2}.$$

**Lemma A.4.** *Let Assumptions 2.2 and 2.3 hold. If the descent inequality*

$$f(x^{k+1}) \leq f(x^k) - \frac{1}{3\sqrt{H_k}}\|\nabla f(x^k)\|^{3/2},$$

*with bounded $H_k \leq H$ holds for all $k \geq 0$, then*

$$f(x^{k+1}) - f_\star \leq \frac{81 H D^3}{k^2}\left(1 + \left(\frac{f(x^0) - f_\star}{9 H D^3}\right)^{1/2}\right)^2.$$

*Proof.* By Assumptions 2.2 and 2.3,

$$f(x^k) - f_\star \leq \langle \nabla f(x^k), x^k - x^\star \rangle \leq \|\nabla f(x^k)\|\|x^k - x^\star\| \leq D\|\nabla f(x^k)\|.$$

Hence $\|\nabla f(x^k)\| \geq \big(f(x^k) - f_\star\big)/D$. Using $H_k \leq H$,

$$f(x^{k+1}) - f_\star \leq f(x^k) - f_\star - \frac{1}{3\sqrt{H_k}}\|\nabla f(x^k)\|^{3/2} \leq f(x^k) - f_\star - \frac{1}{3\sqrt{H}}\|\nabla f(x^k)\|^{3/2}$$

$$\leq f(x^k) - f_\star - \frac{1}{3\sqrt{H}}\Big(\frac{f(x^k) - f_\star}{D}\Big)^{3/2} = f(x^k) - f_\star - \frac{1}{3\sqrt{H D^3}}\big(f(x^k) - f_\star\big)^{3/2}. \quad (20)$$

In (Nesterov, 2022, Lemma A.1) it is shown that if a non–negative sequence $\{\xi_t\}$ satisfies for $\beta > 0$

$$\xi_k - \xi_{k+1} \geq \xi_{k+1}^{1+\beta}$$

then for all $k \geq 0$

$$\xi_k \leq \left[\left(1 + \tfrac{1}{\beta}\right)\left(1 + \xi_0^\beta\right)\tfrac{1}{k}\right]^{1/\beta}. \quad (21)$$

With $\xi_k = \frac{f(x^k) - f_\star}{9 H D^3}$, $\beta = 0.5$ inequality (20) implies

$$f(x^{k+1}) - f_\star \leq \frac{81 H D^3}{k^2}\left(1 + \left(\frac{f(x^0) - f_\star}{9 H D^3}\right)^{1/2}\right)^2.$$

□

**Lemma A.5.** *Let Assumptions 2.2 and 2.3 hold and assume $L_k \leq L$ for all $k$. If for all $k$*

$$f(x^{k+1}) \leq f(x^k) - \frac{8(1-\alpha)\alpha}{9L_k}\|\nabla f(x^k)\|^2,$$

*with bounded $L_k \leq L$ holds for all $k \geq 0$, then*

$$f(x^{k+1}) - f_\star \leq \frac{9LD^2}{8(1-\alpha)\alpha} \cdot \frac{1}{k+1}.$$

*Proof.* By Assumptions 2.2 and 2.3,

$$f(x^k) - f_\star \leq \langle \nabla f(x^k), x^k - x^\star \rangle \leq \|\nabla f(x^k)\|\|x^k - x^\star\| \leq D\|\nabla f(x^k)\|,$$

hence $\|\nabla f(x^k)\|^2 \geq \left(f(x^k) - f_\star\right)^2/D^2$. Using $L_k \leq L$,

$$f(x^{k+1}) - f_\star \leq f(x^k) - f_\star - \frac{8(1-\alpha)\alpha}{9L_k}\|\nabla f(x^k)\|^2 \leq f(x^k) - f_\star - \frac{8(1-\alpha)\alpha}{9LD^2}(f(x^k) - f_\star)^2.$$

Divide both sides of

$$f(x^{k+1}) - f_\star \leq f(x^k) - f_\star - \frac{8(1-\alpha)\alpha}{9LD^2}(f(x^k) - f_\star)^2$$

by $(f(x^{k+1}) - f_\star)(f(x^k) - f_\star) > 0$ and use monotonicity $f(x^{k+1}) \leq f(x^k)$ to get

$$\frac{1}{f(x^{k+1}) - f_\star} - \frac{1}{f(x^k) - f_\star} \geq \frac{8(1-\alpha)\alpha}{9LD^2}.$$

Summing from $t = 0$ to $k$ yields

$$\frac{1}{f(x^{k+1}) - f_\star} \geq \frac{1}{f(x^0) - f_\star} + \frac{8(1-\alpha)\alpha}{9LD^2}(k+1),$$

which gives the stated bound. □

# B MISSING PROOFS FOR CACUN

**Theorem B.1** (Theorem 4.2 duplication). *Let Assumptions 2.1 to 2.3 hold. Then the iterates of Algorithm 1 satisfy the following*

    *1. If $f(x^0) - f_\star \geq 3HD^3$, then*

$$f(x^1) - f_\star \leq 3HD^3.$$

    *2. If $f(x^0) - f_\star \leq 3HD^3$, then for all $k \geq 0$,*

$$f(x^k) - f_\star \leq \frac{3HD^3}{(1 + \frac{k}{3})^2}.$$

*Proof.* For the *gradient* branch we have

$$f(x^{k+1}) \leq f(x^k) - \frac{(2/3)^{3/2}}{\sqrt{2H}} \|\nabla f(x^k)\|^{3/2} \equiv \min_y \big[ f(x^k) + \langle \nabla f(x^k), y - x^k \rangle + H\|y - x^k\|^3 \big]$$

$$\leq \min_y \big[ f(y) + H\|y - x^k\|^3 \big]$$

This is identical to the (Nesterov & Polyak, 2006, Inequality 2.10 with $L = 2H$) used in the CRN analysis, meaning that

$$f(x^{k+1}) \leq \min_y \big[ f(y) + H\|y - x^k\|^3 \big] \tag{22}$$

holds in both cases.

Next, we proceed exactly as in the proof of (Nesterov & Polyak, 2006, Theorem 4).

Indeed, in view of Equation (22), Assumptions 2.2 and 2.3, for any $k \geq 0$ we have

$$f(x^{k+1}) - f_\star \leq \min_y \Big\{ f(y) - f_\star + H\|y - x^k\|^3 : y = \alpha x^* + (1 - \alpha)x^k, \ \alpha \in [0, 1] \Big\}$$

$$\leq \min_{\alpha \in [0,1]} \Big[ f(x^k) - f_\star - \alpha\big(f(x^k) - f_\star\big) + H\alpha^3 \|x^* - x^k\|^3 \Big]$$

$$\leq \min_{\alpha \in [0,1]} \Big[ f(x^k) - f_\star - \alpha\big(f(x^k) - f_\star\big) + H\alpha^3 D^3 \Big].$$

The minimum of the last expression over $\alpha \geq 0$ is attained at

$$\alpha_k = \sqrt{\frac{f(x^k) - f_\star}{3HD^3}}.$$

If $\alpha_k \geq 1$, then the actual optimum over $[0, 1]$ corresponds to $\alpha = 1$, and

$$f(x^{k+1}) - f_\star \leq HD^3.$$

Since the process is monotone, this can happen only at the first iteration.

Assume now that $\alpha_k \leq 1$. Then

$$f(x^{k+1}) - f_\star \leq f(x^k) - f_\star - \left[ \frac{2}{3}\big(f(x^k) - f_\star\big) \right]^{3/2} \frac{1}{\sqrt{2HD^3}}.$$

Equivalently, in the notation $\alpha_k = \sqrt{\frac{f(x^k) - f_\star}{3HD^3}}$

$$\alpha_{k+1}^2 \leq \alpha_k^2 - \frac{2}{3}\alpha_k^3 < \alpha_k^2.$$

Therefore,

$$\frac{1}{\alpha_{k+1}} - \frac{1}{\alpha_k} = \frac{\alpha_k - \alpha_{k+1}}{\alpha_k \alpha_{k+1}} = \frac{\alpha_k^2 - \alpha_{k+1}^2}{\alpha_k \alpha_{k+1}(\alpha_k + \alpha_{k+1})} \geq \frac{\alpha_k^2 - \alpha_{k+1}^2}{2\alpha_k^3} \geq \frac{1}{3}.$$

Thus,

$$\frac{1}{\alpha_k} \geq \frac{1}{\alpha_0} + \frac{k}{3} \geq 1 + \frac{k}{3},$$

and inequality the result

$$f(x^k) - f_\star \leq \frac{3(2H)D^3}{2(1 + \frac{k}{3})^2}, \qquad k \geq 0$$

follows. $\qquad\qquad\square$

## C  MISSING PROOFS FOR ACCELERATED CACUN

Following (Nesterov & Polyak, 2006), define the mapping

$$T_M(x) \overset{\text{def}}{=} \underset{y}{\operatorname{argmin}} \left\{ f(x) + \langle \nabla f(x), y - x \rangle + \frac{1}{2} \langle \nabla^2 f(x)(y - x), y - x \rangle + \frac{M}{3} \|y - x\|^3 \right\}. \quad (23)$$

Then $T = T_M(x)$ is the unique solution of

$$\nabla f(x) + \nabla^2 f(x)(T - x) + M\|T - x\|(T - x) = 0. \quad (24)$$

Denote $r_M(x) = \|x - T_M(x)\|$. Using (24) and the $2H$-Lipschitz Hessian bound Equation (2), we get

$$\|\nabla f(T)\| = \left\| \nabla f(T) - \nabla f(x) - \nabla^2 f(x)(T - x) - Mr_M(x)(T - x) \right\|$$
$$\leq (H + M)r_M(x)^2. \quad (25)$$

**Lemma C.1.** *(Nesterov & Polyak, 2006, Lemma 4)  For any $x \in \mathbb{R}^d$*

$$T_{H_k}(x^k) \leq \min_y \left[ f(y) + \frac{H + H_k}{3} \|y - x^k\|^3 \right]$$

*Proof.* By Assumption 2.1

$$f(y) \leq f(x) + \langle \nabla f(x), y - x \rangle + \frac{1}{2} \langle \nabla^2 f(x)(y - x), y - x \rangle + \frac{H}{3} \|y - x\|^3.$$

The lower bound in the above implies

$$f(x) + \langle \nabla f(x), y - x \rangle + \frac{1}{2} \langle \nabla^2 f(x)(y - x), y - x \rangle \leq f(y) + \frac{H}{3} \|y - x\|^3 \quad (26)$$

Let

$$T_{H_k}(x^k)$$
$$= \underset{y}{\operatorname{argmin}} \left[ f(x^k) + \langle \nabla f(x^k), y - x^k \rangle + \frac{1}{2} \langle \nabla^2 f(x^k)(y - x^k), y - x^k \rangle + \frac{H_k}{3} \|y - x^k\|^3 \right]$$

Then adding $\frac{H_k}{3} \|y - x^k\|^3$ to both sides of (26) and minimizing w.r.t $y$ gives

$$f(T_{H_k}(x^k))$$
$$= \min_y \left[ f(x^k) + \langle \nabla f(x^k), y - x^k \rangle + \frac{1}{2} \langle \nabla^2 f(x^k)(y - x^k), y - x^k \rangle + \frac{M_k}{3} \|y - x^k\|^3 \right]$$
$$\leq \min_y \left[ f(y) + \frac{H + H_k}{3} \|y - x^k\|^3 \right] \quad (27)$$

$\qquad\qquad\square$

**Lemma C.2.** *(Nesterov, 2008, Lemma 6) If $M \geq 2H$, then*

$$\langle \nabla f(T), x - T \rangle \geq \frac{1}{\sqrt{H + M}} \|\nabla f(T)\|^{3/2}. \tag{28}$$

*Proof.* Let $T = T_M(x)$ and Denote $r = r_M(x) = \|x - T_M(x)\|$. Using (2) and (24),

$$H^2 r^4 = \left(H\|T - x\|^2\right)^2 \geq \|\nabla f(T) - \nabla f(x) - \nabla^2 f(x)(T - x)\|^2 = \|\nabla f(T) + Mr(T - x)\|^2,$$

hence

$$\|\nabla f(T)\|^2 + 2Mr\langle \nabla f(T), T - x \rangle + M^2 r^4 \leq H^2 r^4.$$

Rearranging,

$$\langle \nabla f(T), x - T \rangle \geq \frac{1}{2Mr}\|\nabla f(T)\|^2 + \frac{M^2 - H^2}{2M}r^3. \tag{29}$$

For $M \geq 2H$, the derivative in $r$ of the right-hand side of (29) is nonnegative by (25):

$$-\frac{1}{2Mr^2}\|\nabla f(T)\|^2 + \frac{3}{2M}(M^2 - H^2)r^2 \geq -\frac{1}{2Mr^2}\|\nabla f(T)\|^2 + \frac{(H + M)^2}{M}\frac{r^2}{2} \geq 0.$$

Thus, its minimum is attained at the boundary point

$$r = \left(\frac{1}{H + M}\|\nabla f(T)\|\right)^{1/2}.$$

$\square$

Introduce a sequence of estimate functions:

$$\psi_k(x) = \ell_k(x) + \frac{N}{3}\|x - x_0\|^3, \quad k = 1, 2, \ldots$$

where $\ell_k(x)$ are linear functions and $N$ is a positive real parameter.

And a sequence of scaling parameters $\{A_k\}_{k=1}^{\infty}$:

$$A_{k+1} := A_k + a_k \quad k = 1, 2, \ldots$$

**Lemma C.3.** *Let Assumptions 2.1 to 2.3 hold. Then the iterates generated by Algorithm 2 with $M_0 \leq H$ satisfy the following*

$$A_k f(x^k) \leq \psi_k^* \equiv \min_x \psi_k(x), \tag{C1}$$

$$\psi_k(x) \leq A_k f(x) + \frac{2H + N}{3}\|x - x_0\|^3, \quad \forall x \in \mathbb{R}^d, \tag{C2}$$

*where $A_k = \frac{k(k+1)(k+2)}{6}$.*

*Proof.* Let us ensure that relations hold for $k = 1$.

We choose

$$A_1 = 1, \quad \ell_1(x) \equiv f(x^1), x \in \mathbb{R}^d, \tag{30}$$

so

$$\psi_1(x) = f(x^1) + \frac{N}{3}\|x - x_0\|^3. \tag{31}$$

Then $\psi_1^* = f(x^1)$, so Equation (C1) holds for both cases below.

> If $\quad f\left(y^0 - \frac{\nabla f(y^0)}{\sqrt{M_0\|\nabla f(y^0)\|}}\right) \leq f(y^0) - \frac{\sqrt{2}}{3\sqrt{M_0}}\|\nabla f(y^0)\|^{3/2}$ for some $M_0 \leq H$ then we do the
>
> gradient step $x^1 = y^0 - \frac{\nabla f(y^0)}{\sqrt{M_0\|\nabla f(y^0)\|}}$, and have that
>
> $$f(x^1) \leq f(y^0) - \frac{\sqrt{2}}{3\sqrt{M_0}}\|\nabla f(y^0)\|^{3/2}$$
>
> $$\equiv \min_y\left[f(y^0) + \langle \nabla f(y^0), y - y^0 \rangle + \frac{2M_0}{3}\|y - y^0\|^3\right] \leq \min_y\left[f(y) + \frac{2H}{3}\|y - y^0\|^3\right].$$

Else we perform the CRN step $x^1 = T_{2H}(x^0)$, and by Lemma C.1 we have

$$f(x^1) \le \min_y \left[ f(x) + \frac{2H}{3} \|x^1 - x^0\|^3 \right].$$

Thus in both cases

$$\psi_1(x) = f(x^1) + \frac{N}{3} \|x - x_0\|^3 \le \min_y \left[ f(y) + \frac{2H}{3} \|y - x_0\|^3 \right] + \frac{N}{3} \|x - x_0\|^3$$

$$\le f(x) + \frac{2H + N}{3} \|x - x_0\|^3, \quad (32)$$

and (C2) follows.

Assume now that relations (C1),(C2) hold for some $k \ge 1$.

Denote

$$\psi_k(x) \equiv \ell_k(x) + \frac{N}{3} \|x - x_0\|^3 \equiv \ell_k(x) + Nd(x)$$

and

$$v_k = \arg\min_x \psi_k(x).$$

Then the optimality condition reads as

$$\nabla \ell_k(v^k) + N \nabla d(v^k) = 0.$$

Next, applying (Nesterov, 2008, Lemma 4) with $p = 3$, for any $x \in \mathbb{R}^d$, we have

$$d(x) - d(v^k) - \langle \nabla d(v^k), x - v^k \rangle \ge \frac{1}{6} \|x - v^k\|^3,$$

and by convexity

$$\ell_k(x) \ge \ell_k(v^k) + \langle \nabla \ell_k(v^k), x - v^k \rangle.$$

The latter implies

$$\begin{aligned}
\psi_k(x) &\ge \ell_k(v^k) + \langle \nabla \ell_k(v^k), x - v^k \rangle + Nd(v^k) + \langle N \nabla d(v^k), x - v^k \rangle + \frac{N}{6} \|x - v^k\|^3 \\
&= \psi_k(v^k) + \frac{N}{6} \|x - v^k\|^3 \equiv \psi_k^* + \frac{N}{6} \|x - v^k\|^3 \\
&\ge A_k f(x^k) + \frac{N}{6} \|x - v^k\|^3,
\end{aligned}$$

where the last follows from (C1) for $k$.

Let us choose some $a_k > 0$. Define

$$\alpha_k = \frac{a_k}{A_k + a_k},$$

$$y_k = (1 - \alpha_k)x_k + \alpha_k v_k. \quad (33)$$

If $f\left(y^k - \frac{\nabla f(y^k)}{\sqrt{M_k \|\nabla f(y^k)\|}}\right) \le f(y^k) - \frac{1}{\sqrt{3M_k}} \|\nabla f(y^k)\|^{3/2}$ with $M_k \le H$ then the gradient step is performed

$$\begin{aligned}
x_{k+1} &= y^k - \frac{\nabla f(y^k)}{\sqrt{M_k \|\nabla f(y^k)\|}} \\
\psi_{k+1}(x) &= \psi_k(x) + a_k \big[ f(y_k) + \langle \nabla f(y_k), x - y_k \rangle \big].
\end{aligned}$$

In view of (C2) for $k$, for any $x \in \mathbb{R}^d$ we have

$$\psi_{k+1}(x) = \psi_k(x) + a_k\big[f(y_k) + \langle \nabla f(y_k), x - y_k\rangle\big]$$

$$\leq A_k f(x) + \frac{2H + N}{3}\|x - x_0\|^3 + a_k\big[f(y_k) + \langle \nabla f(y_k), x - y_k\rangle\big]$$

$$\leq (A_k + a_k)f(x) + \frac{2H + N}{3}\|x - x_0\|^3,$$

where the last inequality follows from convexity and implies (C2) for $k + 1$.

Next,

$$\psi_{k+1}^* = \min_x \left\{\psi_k(x) + a_k\left[f(y^k) + \langle \nabla f(y^k), x - y^k\rangle\right]\right\}$$

$$\geq \min_x \left\{A_k f(x^k) + \frac{N}{6}\|x - v^k\|^3 + a_k\left[f(y^k) + \langle \nabla f(y^k), x - y^k\rangle\right]\right\}$$

$$\overset{(3)}{\geq} \min_x \left\{A_k(f(y^k) + \langle \nabla f(y^k), x^k - y^k\rangle) + \frac{N}{6}\|x - v^k\|^3 + a_k\left[f(y^k) + \langle \nabla f(y^k), x - y^k\rangle\right]\right\}$$

$$= \min_x \left\{(A_k + a_k)f(y^k) + A_k\langle \nabla f(y^k), x^k - y^k\rangle + a_k\langle \nabla f(y^k), x - y^k\rangle + \frac{N}{6}\|x - v^k\|^3\right\}$$

$$= A_{k+1}f(y^k) + \langle \nabla f(y^k), A_k(x^k - y^k) + a_k(v^k - y^k)\rangle$$

$$+ \min_x \left\{a_k\langle \nabla f(y^k), x - v^k\rangle + \frac{N}{6}\|x - v^k\|^3\right\}$$

$$\overset{(33)}{=} \min_x \left\{A_{k+1}f(x^{k+1}) + \frac{A_{k+1}}{\sqrt{3H}}\|\nabla f(y^k)\|^{3/2} + a_k\langle \nabla f(y^k), x - v^k\rangle + \frac{N}{6}\|x - v^k\|^3\right\}.$$

Hence, our choice of parameters must ensure the following inequality:

$$\frac{A_{k+1}}{\sqrt{3H}}\|\nabla f(y^k)\|^{3/2} + a_k\langle \nabla f(y^k), x - v^k\rangle + \frac{N}{6}\|x - v^k\|^3 \geq 0$$

Using (Nesterov, 2008, Lemma 2) with $p = 3$, $s = a_k\nabla f(y^k)$, and $\sigma = \frac{1}{2}N$, we come to the following condition:

$$A_{k+1}\sqrt{\frac{1}{3H}} \geq \frac{2}{3}\sqrt{\frac{2}{N}}a_k^{3/2}. \tag{34}$$

Else we perform CRN step

$$x_{k+1} = T_{2H}(y_k)$$
$$\psi_{k+1}(x) = \psi_k(x) + a_k\big[f(x_{k+1}) + \langle \nabla f(x_{k+1}), x - x_{k+1}\rangle\big].$$

In view of (C2) for $k$, for any $x \in \mathbb{R}^d$ we have

$$\psi_{k+1}(x) = \psi_k(x) + a_k\big[f(x_{k+1}) + \langle \nabla f(x_{k+1}), x - x_{k+1}\rangle\big]$$

$$\leq A_k f(x) + \frac{2H + N}{3}\|x - x_0\|^3 + a_k\big[f(x_{k+1}) + \langle \nabla f(x_{k+1}), x - x_{k+1}\rangle\big]$$

$$\leq (A_k + a_k)f(x) + \frac{2H + N}{3}\|x - x_0\|^3,$$

we the last inequality follows from convexity and implies (C2) for $k + 1$.

Next,

$$\psi_{k+1}^* = \min_x \left\{ \psi_k(x) + a_k \left[ f(x^{k+1}) + \langle \nabla f(x^{k+1}), x - x^{k+1} \rangle \right] \right\}$$

$$\geq \min_x \left\{ A_k f(x^k) + \frac{N}{6} \|x - v^k\|^3 + a_k \left[ f(x^{k+1}) + \langle \nabla f(x^{k+1}), x - x^{k+1} \rangle \right] \right\}$$

$$\geq \min_x \left\{ (A_k + a_k) f(x^{k+1}) + A_k \langle \nabla f(x^{k+1}), x^k - x^{k+1} \rangle + a_k \langle \nabla f(x^{k+1}), x - x^{k+1} \rangle + \frac{N}{6} \|x - v^k\|^3 \right\}$$

$$= A_{k+1} f(x^{k+1}) + \langle \nabla f(x^{k+1}), A_{k+1} y^k - a_k v^k - A_k x^{k+1} \rangle$$

$$+ \min_x \left\{ a_k \langle \nabla f(x^{k+1}), x - x^{k+1} \rangle + \frac{N}{6} \|x - v^k\|^3 \right\}$$

$$= \min_x \left\{ A_{k+1} f(x^{k+1}) + A_{k+1} \langle \nabla f(x^{k+1}), y^k - x^{k+1} \rangle + a_k \langle \nabla f(x^{k+1}), x - v^k \rangle + \frac{N}{6} \|x - v^k\|^3 \right\}.$$

Further, by Lemma C.2 we have

$$\langle \nabla f(x^{k+1}), y^k - x^{k+1} \rangle \geq \sqrt{\frac{1}{M+H}} \|\nabla f(x^{k+1})\|^{3/2}.$$

Hence, our choice of parameters must ensure the following inequality:

$$A_{k+1} \sqrt{\frac{1}{3H}} \|\nabla f(x^{k+1})\|^{3/2} + a_k \langle \nabla f(x^{k+1}), x - v^k \rangle + \frac{N}{6} \|x - v^k\|^3 \geq 0, \quad \forall x \in \mathbb{R}^d.$$

Using (Nesterov, 2008, Lemma 2) with $p = 3$, $s = a_k \nabla f(x^{k+1})$, and $\sigma = \frac{1}{2} N$, we come to the following condition:

$$A_{k+1} \sqrt{\frac{1}{3H}} \geq \frac{2}{3} \sqrt{\frac{2}{N}} a_k^{3/2}. \tag{35}$$

In both cases one can satisfy the conditions by choosing

$$A_k = \frac{k(k+1)(k+2)}{6},$$

$$a_k = A_{k+1} - A_k = \frac{(k+1)(k+2)(k+3)}{6} - \frac{k(k+1)(k+2)}{6} = \frac{(k+1)(k+2)}{2}$$

for $k \geq 1$. Since

$$a_k^{-3/2} A_{k+1} = 2^{3/2} \frac{(k+1)(k+2)(k+3)}{6[(k+1)(k+2)]^{3/2}} = 2^{1/2} \frac{k+3}{3[(k+1)(k+2)]^{1/2}} \geq \frac{\sqrt{2}}{3},$$

inequalities (34) and (35) lead to the following condition:

$$\frac{1}{\sqrt{3H}} \geq \frac{2}{\sqrt{N}}.$$

Hence one can chose

$$N = 12H.$$

$\square$

**Theorem C.4** (Theorem 4.3 duplication). *Let Assumptions 2.1 to 2.3 hold. Then the iterates generated by Algorithm 2 with $M_0 \leq H$ satisfy the following*

$$f(x^k) - f(x^*) \leq \frac{14 \cdot 2H \|x^0 - x^*\|^3}{k(k+1)(k+2)}.$$

*Proof.* Indeed, we have shown that

$$A_k f(x^k) \overset{\text{(C1)}}{\leq} \psi_k^* \overset{\text{(C2)}}{\leq} A_k f(x^*) + \frac{2H + N}{3} \|x^0 - x^*\|^3.$$

The theorem claim follows form rearranging and $A_k$ expression from Lemma C.3.

$\square$

*Remark* C.5. Note that the point $v^k$ can be found in (4.8) by an explicit formula. Consider

$$s^k = \nabla \ell_k(x).$$

This vector does not depend on $x$ since the function $\ell_k(x)$ is linear. Then

$$v^k = x^0 - \frac{s^k}{\sqrt{N\|s^k\|}}.$$

# D  CASUAL CUBIC ADAN

Same technique is also applicable for regularized global Newton (Mishchenko, 2023), which we refer as RegNewton. We simply perform

$$x^{k+1} = x^k - \frac{\nabla f(x^k)}{\sqrt{H_k \|\nabla f(x^k)\|}} \tag{36}$$

if

$$f\left(x^k - \frac{\nabla f(x^k)}{\sqrt{H_k \|\nabla f(x^k)\|}}\right) \le f(x^k) - \frac{2}{3 \cdot 64 \sqrt{H_k}} \|\nabla f(x^k)\|^{3/2}, \tag{37}$$

where, for the non-adaptive RegNewton, $H_k \equiv H$ is fixed; $H$ is a known global upper bound on the Hessian–Lipschitz constant.

Instead of assuming a known global $H$, we adapt a local Hessian–Lipschitz estimate $H_k$ from first-order information via (4) and accept the gradient step (36) whenever the AdaN (Mishchenko, 2023) descent test (37) holds, which is the key inequality for AdaN convergence. The resulting scheme does not always improve wall-clock time. But we observe consistent gains with a simple two-stage scheme: run the gradient step governed by (37) (backtracking on $H_k$) until the test first fails, then switch to AdaN. This also "warms up" $H_k$ using only first-order backtracking. The resulting algorithm is listed as Algorithm 4.

---

**Algorithm 4** Casual Cubic AdaN (CaCuAdaN)

---

1: **Input:** $x^0 \in \mathbb{R}^d$, $H > 0$, $H > 0$, $flag = 0$
2: **for** $k = 0, 1, \ldots$ **do**
3:     **if not** *flag* **then**
4:         $H = H/2$
5:         **while** $f\left(x^k - \frac{\nabla f(x^k)}{\sqrt{H\|\nabla f(x^k)\|}}\right) \ge f(x^k) + \frac{1}{2}\frac{\langle \nabla f(x^k), \nabla^2 f(x^k)\nabla f(x^k)\rangle}{H\|\nabla f(x^k)\|} - \frac{2}{3\sqrt{H}}\|\nabla f(x^k)\|^{3/2}$ **do**
6:             $H = 2H$
7:         **if** $f\left(x^k - \frac{\nabla f(x^k)}{\sqrt{H\|\nabla f(x^k)\|}}\right) \ge f(x^k) - \frac{2}{3\cdot 64\sqrt{H}}\|\nabla f(x^k)\|^{3/2}$ **then**
8:             *flag = 1*
9:             **break**
10:         $x^{k+1} = x^k - \frac{\nabla f(x^k)}{\sqrt{H\|\nabla f(x^k)\|}}$
11:     **else**
12:         $x^{k+1}$ is calculated by AdaN step.

---

Analogously to AdaN+ (Mishchenko, 2023), we propose a heuristic extension that uses gradient-based backtracking to calibrate the local Hessian–Lipschitz constant. The resulting algorithm is listed as Algorithm 5.

---

**Algorithm 5** Casual Cubic AdaN+ (CaCuAdaN+)

---

1: **Input:** $x^0 \in \mathbb{R}^d$, $H > 0$,
2: **for** $k = 0, 1, \ldots$ **do**
3:     $H = H/8$
4:     **while** $f\left(x^k - \frac{\nabla f(x^k)}{\sqrt{H\|\nabla f(x^k)\|}}\right) \ge f(x^k) + \frac{1}{2}\frac{\langle \nabla f(x^k), \nabla^2 f(x^k)\nabla f(x^k)\rangle}{H\|\nabla f(x^k)\|} - \frac{2}{3\sqrt{H}}\|\nabla f(x^k)\|^{3/2}$ **do**
5:         $H = 2H$
6:     $x^{k+1} = x^k - (\nabla^2 f(x^k) + \sqrt{H\|\nabla f(x^k)\|}\mathbf{I})^{-1}\nabla f(x^k),$

---

# E  MISSING PROOFS FOR FIRST ORDER RESULTS

## E.1  EXAMPLES

**Example E.1** (Example 5.1 duplication)**.** *Let $f(x) = \frac{H}{3}\|x\|^3$ with $H > 0$. Then $\nabla^2 f$ is $2H$-Lipschitz and* (10) *holds with $C = 4H$.*

*Proof.* Write $r = \|x\|$ and $e = x/r$ for $x \neq 0$. Then

$$\nabla f(x) = Hrx, \qquad \nabla^2 f(x) = H\Big(rI + \frac{xx^\top}{r}\Big) = Hr(I + ee^\top).$$

(At $x = 0$, set $\nabla^2 f(0) = 0$, which is the continuous extension since $\|xx^\top/r\| = r \to 0$.)

*Verification of* (10)*.* Using the formulas above,

$$\nabla^2 f(x)\nabla f(x) = H\Big(rI + \frac{xx^\top}{r}\Big)(Hrx) = 2H^2 r^2 x,$$

hence

$$\frac{\langle \nabla f(x), \nabla^2 f(x)\nabla f(x)\rangle}{\|\nabla f(x)\|^2} = \frac{(Hrx)^\top(2H^2 r^2 x)}{\|Hrx\|^2} = 2Hr.$$

Since $\|\nabla f(x)\| = Hr^2$, we get

$$2Hr = 2\sqrt{H}\sqrt{\|\nabla f(x)\|} = \sqrt{(4H)\|\nabla f(x)\|},$$

so (10) holds with $C = 4H$ (with equality for $x \neq 0$).

**Lipschitzness of the Hessian.** Fix a unit vector $u$ and define

$$\phi(t) := u^\top \nabla^2 f\big(y + t(x - y)\big)u, \quad t \in [0, 1].$$

Let $z(t) = y + t(x - y)$, $r = \|z\|$, $\alpha = \langle u, z\rangle$, and $c = \alpha/r$ (define by continuity at $r = 0$). Then

$$\phi(t) = H\left(r + \frac{\alpha^2}{r}\right), \quad \phi'(t) = H\Big(r'(1 - c^2) + 2c\alpha'\Big),$$

with $r' = \langle z/r, x - y\rangle$ and $\alpha' = \langle u, x - y\rangle$. Thus $|r'| \leq \|x - y\|$, $|\alpha'| \leq \|x - y\|$, and $|c| \leq 1$, giving

$$|\phi'(t)| \leq H\big((1 - c^2) + 2|c|\big)\|x - y\| \leq 2H\|x - y\|.$$

By the mean value theorem,

$$|u^\top(\nabla^2 f(x) - \nabla^2 f(y))u| = |\phi(1) - \phi(0)| \leq \int_0^1 |\phi'(t)|dt \leq 2H\|x - y\|.$$

Taking the supremum over unit $u$ yields $\|\nabla^2 f(x) - \nabla^2 f(y)\| \leq 2H\|x - y\|$. $\qquad\square$

**Example E.2** (Example 5.2 duplication)**.** *Let $f(x) = \sum_{i=1}^d \frac{H_i}{3}|x_i|^3$ with $H_i > 0$ and $H_{\max} = \max_i H_i$. Then $\nabla^2 f$ is $2H_{\max}$-Lipschitz and* (10) *holds with $C = 4H_{\max}$.*

*Proof. Formulas.* $\nabla f(x) = (H_i x_i |x_i|)_i, \qquad \nabla^2 f(x) = \text{diag}(2H_i|x_i|).$

*Inequality* (10)*.* Since $\nabla^2 f(x)$ is diagonal with entries $2H_i|x_i|$,

$$\frac{\langle \nabla f(x), \nabla^2 f(x)\nabla f(x)\rangle}{\|\nabla f(x)\|^2} = \frac{\sum_i 2H_i|x_i|(H_i x_i|x_i|)^2}{\sum_i(H_i x_i|x_i|)^2} \leq \max_i 2H_i|x_i|.$$

Let $\alpha_i := H_i x_i^2 \geq 0$. Then $\|\nabla f(x)\| = (\sum_i \alpha_i^2)^{1/2}$ and

$$\max_i 2H_i|x_i| = 2\max_i \sqrt{H_i}\sqrt{\alpha_i} \leq 2\sqrt{H_{\max}}\max_i \sqrt{\alpha_i} \leq 2\sqrt{H_{\max}}(\sum_i \alpha_i^2)^{1/4}$$

$$= 2\sqrt{H_{\max}}\|\nabla f(x)\|^{1/2}.$$

Thus (10) holds with $C = 4H_{\max}$.

**Lipschitzness of the Hessian.** For any $x, y$,

$$\|\nabla^2 f(x) - \nabla^2 f(y)\| = \Big\|\text{diag}\big(2H_i(|x_i| - |y_i|)\big)\Big\| \leq \max_i 2H_i|x_i - y_i| \leq 2H_{\max}\|x - y\|.$$

$$\square$$

**Example E.3** (Example 5.3 duplication). *Let $a \in \mathbb{R}^d \setminus \{0\}$ and $f(x) = \log\left(1 + e^{-a^\top x}\right)$. Then $\nabla^2 f$ is $\frac{\|a\|^3}{6\sqrt{3}}$-Lipschitz and* (10) *holds with $C = \frac{4}{27}\|a\|^3$.*

*Proof.* Set $s = a^\top x$ and $\sigma(t) = \frac{1}{1+e^{-t}}$. Using $\sigma(s) + \sigma(-s) = 1$,

$$\nabla f(x) = \frac{-e^{-s}}{1 + e^{-s}}a = -\sigma(-s)a,$$

and

$$\nabla^2 f(x) = \sigma(s)\sigma(-s)aa^\top.$$

*Inequality* (10). We have

$$\frac{\langle \nabla f, \nabla^2 f \nabla f \rangle}{\|\nabla f\|^2} = \sigma(s)\sigma(-s)\|a\|^2, \qquad \|\nabla f(x)\| = \sigma(-s)\|a\|.$$

Thus (10) is equivalent to

$$\sigma(s)\sigma(-s)\|a\|^2 \leq \sqrt{C\sigma(-s)\|a\|} \quad \text{or} \quad C \geq \sigma(s)^2\sigma(-s)\|a\|^3.$$

Let $p = \sigma(s) \in (0,1)$. Maximizing $p^2(1-p)$ on $[0,1]$ gives $\max_p p^2(1-p) = 4/27$ at $p = 2/3$, hence $C = \frac{4}{27}\|a\|^3$.

**Lipschitzness of the Hessian.** With $\phi(s) = \sigma(s)\sigma(-s)$ one obtains

$$\|\nabla^2 f(x) - \nabla^2 f(y)\| = \|a\|^2|\phi(a^\top x) - \phi(a^\top y)| \leq \|a\|^2\left(\sup_s |\phi'(s)|\right)|a^\top(x-y)| \leq \frac{\|a\|^3}{6\sqrt{3}}\|x-y\|,$$

since $\phi'(s) = \sigma(s)\sigma(-s)(1 - 2\sigma(s))$ and $\max_{p \in [0,1]} p(1-p)|1 - 2p| = 1/(6\sqrt{3})$, while $|a^\top(x - y)| \leq \|a\|\|x - y\|$. $\qquad\square$

### E.2 MISSING PROOFS FOR CACUADGD

### E.3 LIPSCHITZ CASE

**Lemma E.4** (Lemma 5.6 duplication). *Let Assumptions 2.1 to 2.3 and 5.5 hold. Then the iterates generated by Algorithm 3 with $\alpha = 0.5$ satisfy the following*

$$
(i) \quad \left(f(x^k) - f_\star\right)^{1/2} \geq \frac{3}{2}\frac{LD^2}{\sqrt{HD^3}}, \quad then \quad f(x^{k+1}) - f_\star \leq f(x^k) - f_\star - \frac{1}{3\sqrt{HD^3}}\left(f(x^k) - f_\star\right)^{3/2} \quad (38)
$$

$$
(ii) \quad \left(f(x^k) - f_\star\right)^{1/2} \leq \frac{3}{2}\frac{LD^2}{\sqrt{HD^3}}, \quad then \quad f(x^{k+1}) - f_\star \leq f(x^k) - f_\star - \frac{2}{9LD^2}\left(f(x^k) - f_\star\right)^2. \quad (39)
$$

*Proof.* By inequalities (6) and (5) either

$$
f(x^{k+1}) \leq f(x^k) - \frac{1}{3\sqrt{H_k}}\|\nabla f(x^k)\|^{3/2} \quad (H_k \leq H),
$$

or

$$
f(x^{k+1}) \leq f(x^k) - \frac{2}{9L_k}\|\nabla f(x^k)\|^2 \quad (L_k \leq L).
$$

By convexity,

$$
f(x^k) - f_\star \leq \langle \nabla f(x^k), x^k - x^\star \rangle \leq \|\nabla f(x^k)\|\|x^k - x^\star\| \leq D\|\nabla f(x^k)\|.
$$

Hence $\|\nabla f(x^k)\| \geq \left(f(x^k) - f_\star\right)/D$. Using $H_k \leq H$ and $L_k \leq L$,

$$
f(x^{k+1}) \leq f(x^k) - \frac{1}{3\sqrt{H}}\left(\frac{f(x^k) - f_\star}{D}\right)^{3/2} = f(x^k) - \frac{1}{3\sqrt{HD^3}}\left(f(x^k) - f_\star\right)^{3/2},
$$

$$
f(x^{k+1}) \leq f(x^k) - \frac{2}{9L}\left(\frac{f(x^k) - f_\star}{D}\right)^2 = f(x^k) - \frac{2}{9LD^2}\left(f(x^k) - f_\star\right)^2.
$$

The threshold $\left(f(x^k) - f_\star\right)^{1/2} = \frac{3}{2}\frac{LD^2}{\sqrt{HD^3}}$ is the unique solution of

$$
\frac{2}{9LD^2}\left(f(x^k) - f_\star\right)^2 = \frac{1}{3\sqrt{HD^3}}\left(f(x^k) - f_\star\right)^{3/2},
$$

which determines which of the two decreases is smaller; the stated cases follow, as does the definition of $K_1$.

**Detailed derivation.** If $(f(x^k) - f_\star)^{1/2} \geq \frac{3}{2}\frac{LD^2}{\sqrt{HD^3}}$ then

$$
f(x^{k+1}) \leq f(x^k) - \frac{2}{9L_k}\|\nabla f(x^k)\|^2 \leq f(x^k) - \frac{2}{9L}\|\nabla f(x^k)\|^2 \leq f(x^k) - \frac{2}{9LD^2}\left(f(x^k) - f_\star\right)^2
$$

$$
\leq f(x^k) - \frac{1}{3\sqrt{HD^3}}\left(f(x^k) - f_\star\right)^{3/2}
$$

coincides with

$$
f(x^{k+1}) \leq f(x^k) - \frac{1}{3\sqrt{H_k}}\|\nabla f(x^k)\|^{3/2} \leq f(x^k) - \frac{1}{3\sqrt{H}}\|\nabla f(x^k)\|^{3/2}
$$

$$
\leq f(x^k) - \frac{1}{3\sqrt{HD^3}}\left(f(x^k) - f_\star\right)^{3/2}.
$$

This implies that

$$
f(x^{k+1}) \leq f(x^k) - \frac{1}{3\sqrt{HD^3}}\left(f(x^k) - f_\star\right)^{3/2}
$$

for any value of $M_k$.

Similarly if $(f(x^k) - f_\star)^{1/2} \leq \frac{3}{2} \frac{LD^2}{\sqrt{HD^3}}$ then

$$f(x^{k+1}) \leq f(x^k) - \frac{2}{9L_k} \|\nabla f(x^k)\|^2 \leq f(x^k) - \frac{2}{9L} \|\nabla f(x^k)\|^2 \leq f(x^k) - \frac{2}{9LD^2} (f(x^k) - f_\star)^2$$

and

$$f(x^{k+1}) \leq f(x^k) - \frac{1}{3\sqrt{H_k}} \|\nabla f(x^k)\|^{3/2} \leq f(x^k) - \frac{1}{3\sqrt{H}} \|\nabla f(x^k)\|^{3/2}$$
$$\leq f(x^k) - \frac{1}{3\sqrt{HD^3}} (f(x^k) - f_\star)^{3/2} \leq f(x^k) - \frac{2}{9LD^2} (f(x^k) - f_\star)^2.$$

Thus for any $M_k$

$$f(x^{k+1}) \leq f(x^k) - \frac{2}{9LD^2} (f(x^k) - f_\star)^2.$$

$\square$

**Corollary E.5** (Corollary 5.7 duplication). *Let Assumptions 2.1 to 2.3 and 5.5 hold. Then the iterates generated by Algorithm 3 with $\alpha = 0.5$ satisfy $\forall k : (f(x^k) - f_\star)^{1/2} \geq \frac{3}{2} \frac{LD^2}{\sqrt{HD^3}}$ the following*

$$f(x^{k+1}) - f_\star \leq \frac{\left(9\sqrt{HD^3} + 3\sqrt{f(x^0) - f_\star}\right)^2}{k^2}.$$

*Proof.* In (Nesterov, 2022, Lemma A.1) it is shown that if a non–negative sequence $\{\xi_t\}$ satisfies for $\beta > 0$

$$\xi_k - \xi_{k+1} \geq \xi_{k+1}^{1+\beta}$$

then for all $k \geq 0$

$$\xi_k \leq \left[\left(1 + \frac{1}{\beta}\right)\left(1 + \xi_0^\beta\right)\frac{1}{k}\right]^{1/\beta}. \tag{40}$$

With $\xi_k = \frac{f(x^k) - f_\star}{9HD^3}$, $\beta = 0.5$ Inequality (38) implies

$$f(x^{k+1}) - f_\star \leq \frac{81HD^3}{k^2} \left(1 + \left(\frac{f(x^0) - f_\star}{9HD^3}\right)^{1/2}\right)^2.$$

$\square$

**Theorem E.6** (Theorem 5.8 duplication). *Let Assumptions 2.1 to 2.3 and 5.5 hold. Then the iterates generated by Algorithm 3 with $\alpha = 0.5$ satisfy for all $k \geq 0$ the following*

$$f_k - f_* \leq \frac{9LD^2}{k + LD^2\Phi(\xi_0) - \frac{2HD}{L}}, \tag{41}$$

*where $\Phi(\xi) := \frac{9}{2}\frac{1}{\xi} + \frac{6\sqrt{H}}{L\sqrt{D}}\frac{1}{\sqrt{\xi}}$.*

*Or when the algorithm reaches non-accelerated regime $((f(x^k) - f_\star)^{1/2} \leq \frac{3}{2}\frac{LD^2}{\sqrt{HD^3}})$*

$$\xi_k \leq \frac{27}{2} \frac{LD^2}{k + LD^2\Phi(\xi_0)}.$$

*Proof.* Let $\xi_k := f(x^k) - f_\star$. The inequalities (38) and (39) imply

$$\xi_k - \xi_{k+1} > 0,$$

or $x^k$ is a solution.

For any $u > v > 0$,

$$\frac{1}{v} - \frac{1}{u} \geq \frac{u-v}{u^2}, \qquad \frac{1}{\sqrt{v}} - \frac{1}{\sqrt{u}} \geq \frac{u-v}{2u^{3/2}}.$$

With $u = \xi_k$, $v = \xi_{k+1}$,

$$\Phi(\xi_{k+1}) - \Phi(\xi_k) \geq \frac{9}{2} \frac{\xi_k - \xi_{k+1}}{\xi_k^2} + \frac{6\sqrt{H}}{L\sqrt{D}} \cdot \frac{\xi_k - \xi_{k+1}}{2\xi_k^{3/2}}.$$

Consider the two cases:

*(i)* if $\xi_k - \xi_{k+1} \geq \frac{2}{9LD^2}\xi_k^2$, then

$$\Phi(\xi_{k+1}) - \Phi(\xi_k) \geq \frac{9}{2} \cdot \frac{2}{9LD^2} = \frac{1}{LD^2}.$$

*(ii)* if $\xi_k - \xi_{k+1} \geq \frac{1}{3\sqrt{HD^3}}\xi_k^{3/2}$ , then

$$\Phi(\xi_{k+1}) - \Phi(\xi_k) \geq \frac{6\sqrt{H}}{L\sqrt{D}} \cdot \frac{1}{2} \cdot \frac{1}{3\sqrt{H}D^{3/2}} = \frac{1}{LD^2}.$$

So in all cases,

$$\Phi(\xi_k) \geq \Phi(\xi_0) + \frac{k}{LD^2}. \tag{42}$$

To turn (42) into an upper bound on $\xi_k$, use for any $\beta > 0$ and any $\xi > 0$:

$$\frac{6\sqrt{H}}{L\sqrt{D}} \cdot \frac{1}{\sqrt{\xi}} \leq \frac{\beta}{\xi} + \frac{1}{4\beta}\left(\frac{6\sqrt{H}}{L\sqrt{D}}\right)^2 = \frac{\beta}{\xi} + \frac{9H}{\beta L^2 D}. \tag{43}$$

Hence

$$\Phi(\xi_k) = \frac{9}{2}\frac{1}{\xi_k} + \frac{6\sqrt{H}}{L\sqrt{D}}\frac{1}{\sqrt{\xi_k}} \leq \frac{\frac{9}{2} + \beta}{\xi_k} + \frac{9H}{\beta L^2 D}.$$

Combine with (42) and rearrange:

$$\Phi(\xi_0) + \frac{k}{LD^2} \leq \frac{\frac{9}{2} + \beta}{\xi_k} + \frac{9H}{\beta L^2 D} \quad \text{or} \quad \xi_k \leq \frac{\left(\frac{9}{2} + \beta\right)}{\Phi(\xi_0) + \frac{k}{LD^2} - \frac{9H}{\beta L^2 D}},$$

which is (41) (with $\beta = 9/2$).

Assume we are in the non-accelerated region, i.e.

$$\xi_k \leq \frac{9}{4}\frac{L^2 D}{H} \quad \text{or} \quad \frac{H}{L^2 D} \leq \frac{9}{4}\frac{1}{\xi_k}.$$

Taking square roots (all quantities are positive),

$$\frac{\sqrt{H}}{L\sqrt{D}} \leq \frac{3}{2}\frac{1}{\sqrt{\xi_k}}.$$

Hence, we get for the potential

$$\Phi(\xi_k) = \frac{9}{2}\frac{1}{\xi_k} + \frac{6\sqrt{H}}{L\sqrt{D}}\frac{1}{\sqrt{\xi_k}} \leq \frac{9}{2}\frac{1}{\xi_k} + 6 \cdot \frac{3}{2} \cdot \frac{1}{\xi_k} = \frac{27}{2}\frac{1}{\xi_k}.$$

Combining with the telescoping one has

$$\Phi(\xi_k) \geq \Phi(\xi_0) + \frac{k}{LD^2},$$

and finally obtains

$$\Phi(\xi_0) + \frac{k}{LD^2} \leq \Phi(\xi_k) \leq \frac{27}{2}\frac{1}{\xi_k} \quad \text{or} \quad \xi_k \leq \frac{27}{2}\frac{LD^2}{k + LD^2\Phi(\xi_0)}.$$

$\square$

### E.4 RELAXED SMOOTHNESS

**Lemma E.7.** *For all $s \in \mathbb{R}$,*

$$\sigma(s)\sigma(-s) \leq \frac{2}{3\sqrt{3}}\sqrt{\log\left(1 + e^{-s}\right)},$$

*where $\sigma(s) = \frac{1}{1+e^{-s}}$*

*Proof.* Let $t = e^{-s} > 0$. Then

$$\sigma(s)\sigma(-s) = \frac{t}{(1+t)^2}, \qquad \log\left(1 + e^{-s}\right) = \log(1 + t).$$

We use the elementary inequality

$$\log(1 + t) \geq \frac{t}{1+t} \qquad \text{for all } t > 0,$$

which follows from the convexity of $-\log$ or by defining $\phi(t) = \log(1+t) - \frac{t}{1+t}$ and noting $\phi'(t) = \frac{t}{(1+t)^2} \geq 0$ with $\phi(0) = 0$. Hence

$$\frac{\sigma(s)\sigma(-s)}{\sqrt{\log(1+t)}} = \frac{t}{(1+t)^2\sqrt{\log(1+t)}} \leq \frac{t}{(1+t)^2} \cdot \frac{1}{\sqrt{t/(1+t)}} = \frac{\sqrt{t}}{(1+t)^{3/2}} \leq \frac{2}{3\sqrt{3}}.$$

Therefore,

$$\sigma(s)\sigma(-s) \leq \frac{2}{3\sqrt{3}}\sqrt{\log(1+t)} = \frac{2}{3\sqrt{3}}\sqrt{\log\left(1 + e^{-s}\right)}.$$

$\square$

**Example E.8** (Example 5.13 duplication). *Let $a \in \mathbb{R}^d \setminus \{0\}$, $s = \langle a, x \rangle$, and $f(x) = \log(1 + e^{-s})$. Then $f_\star = 0$ and for all $x \in \mathbb{R}^d$,*

$$\frac{\langle \nabla f(x), \nabla^2 f(x)\nabla f(x)\rangle}{\|\nabla f(x)\|^2} \leq \frac{2}{3\sqrt{3}}\|a\|^2\left(f(x) - f_\star\right)^{1/2}.$$

*Proof.* Write $\sigma(t) = \frac{1}{1+e^{-t}}$. Then

$$\nabla f(x) = -a\sigma(-s), \qquad \nabla^2 f(x) = aa^\top\sigma(s)\sigma(-s),$$

and hence

$$\frac{\langle \nabla f, \nabla^2 f \nabla f\rangle}{\|\nabla f\|^2} = \frac{\|a\|^4\sigma(s)\sigma(-s)^3}{\|a\|^2\sigma(-s)^2} = \|a\|^2\sigma(s)\sigma(-s).$$

By Lemma E.7

$$\frac{\langle \nabla f, \nabla^2 f \nabla f\rangle}{\|\nabla f\|^2} \leq \frac{2}{3\sqrt{3}}\|a\|^2\sqrt{f(x)} = \frac{2}{3\sqrt{3}}\|a\|^2\sqrt{f(x) - f_\star},$$

$\square$

where we used that $f_\star = 0$.

**Lemma E.9** (Zero infimum under linear separability). *If there exists $v \in \mathbb{R}^d$ and $\gamma > 0$ such that $y_i\langle a_i, v\rangle \geq \gamma$ for all $i$, then for $f(x) = \frac{1}{n}\sum_{i=1}^n \log\left(1 + e^{-y_i\langle a_i, x\rangle}\right)$ one has $f_\star = 0$.*

*Proof.* For any $\tau > 0$,

$$f(\tau v) = \frac{1}{n}\sum_{i=1}^n \log\left(1 + e^{-\tau y_i\langle a_i, v\rangle}\right) \leq \log\left(1 + e^{-\tau\gamma}\right) \xrightarrow[\tau \to \infty]{} 0.$$

Hence $\inf_x f(x) = 0$. $\square$

**Example E.10** (Example 5.14 duplication). *Let there exists $v \in \mathbb{R}^d$ and $\gamma > 0$ such that $y_i \langle a_i, v \rangle \geq \gamma$ for all $i$, then for $f(x) = \frac{1}{n} \sum_{i=1}^n \log(1 + e^{-y_i \langle a_i, x \rangle})$ one has $f_\star = 0$ and for all $x \in \mathbb{R}^d$,*

$$\frac{\langle \nabla f(x), \nabla^2 f(x) \nabla f(x) \rangle}{\|\nabla f(x)\|^2} \leq \frac{2}{3\sqrt{3}} \sqrt{\frac{1}{n} \sum_{i=1}^n \|a_i\|^4} (f(x) - f_\star)^{1/2}.$$

*Proof.* If $\nabla f(x) = 0$, the left-hand side is $0$ and the claim holds. Hence assume $\nabla f(x) \neq 0$. With $\sigma(t) = \frac{1}{1+e^{-t}}$

$$w_i := \sigma(s_i)\sigma(-s_i), \qquad s_i = y_i \langle a_i, x \rangle.$$
$$\nabla f(x) = -\frac{1}{n} \sum_{i=1}^n y_i a_i \sigma(-s_i), \qquad \nabla^2 f(x) = \frac{1}{n} \sum_{i=1}^n w_i a_i a_i^\top,$$

Using the quadratic form of $\nabla^2 f(x)$

$$\frac{\langle \nabla f(x), \nabla^2 f(x) \nabla f(x) \rangle}{\|\nabla f(x)\|^2} = \frac{\langle \nabla f(x), \left(\frac{1}{n} \sum_{i=1}^n w_i a_i a_i^\top\right) \nabla f(x) \rangle}{\|\nabla f(x)\|^2}$$
$$= \frac{1}{n} \sum_{i=1}^n w_i \frac{(\langle a_i, \nabla f(x) \rangle)^2}{\|\nabla f(x)\|^2}.$$

For each $i$, $(\langle a_i, \nabla f(x) \rangle)^2 \leq \|a_i\|^2 \|\nabla f(x)\|^2$ by Cauchy–Schwarz implying

$$\frac{\langle \nabla f(x), \nabla^2 f(x) \nabla f(x) \rangle}{\|\nabla f(x)\|^2} \leq \frac{1}{n} \sum_{i=1}^n w_i \|a_i\|^2.$$

By Lemma E.7, $w_i = \sigma(s_i)\sigma(-s_i) \leq \frac{2}{3\sqrt{3}} \sqrt{\log(1 + e^{-s_i})}$. Therefore,

$$\frac{1}{n} \sum_{i=1}^n w_i \|a_i\|^2 \leq \frac{2}{3\sqrt{3}} \cdot \frac{1}{n} \sum_{i=1}^n \|a_i\|^2 \sqrt{\log(1 + e^{-s_i})}$$
$$\leq \frac{2}{3\sqrt{3}} \left(\frac{1}{n} \sum_{i=1}^n \|a_i\|^4\right)^{1/2} \left(\frac{1}{n} \sum_{i=1}^n \log(1 + e^{-s_i})\right)^{1/2}$$
$$= \frac{2}{3\sqrt{3}} \left(\frac{1}{n} \sum_{i=1}^n \|a_i\|^4\right)^{1/2} \sqrt{f(x)}.$$

Under linear separability, $f_\star = 0$ (see Lemma E.9). Thus $\sqrt{f(x)} = \sqrt{f(x) - f_\star}$, and combining steps above yields

$$\frac{\langle \nabla f(x), \nabla^2 f(x) \nabla f(x) \rangle}{\|\nabla f(x)\|^2} \leq \frac{2}{3\sqrt{3}} \sqrt{\frac{1}{n} \sum_{i=1}^n \|a_i\|^4} \sqrt{f(x) - f_\star}.$$

$\square$

**Corollary E.11** (Corollary 5.15 duplication). *Let Assumptions 2.2, 2.3, 5.5 and 5.11 hold. Then the total number of iterations needed to reach $f(x^K) - f_\star \leq \varepsilon$ satisfies*

$$K = \mathcal{O}\left(\frac{L_0 D^2 + (f_0 - f_*)}{\varepsilon} + \frac{\left(L_1 D^2 + \sqrt{H D^3} + \sqrt{f_0 - f_*}\right)^2}{\sqrt{\varepsilon}}\right).$$

*Proof.* We begin with bounding the following term and making it non-positive.

$$\langle \nabla f(x^k), \nabla^2 f(x^k) \nabla f(x^k) \rangle - \frac{2\sqrt{M_k}}{3} \|\nabla f(x^k)\|^{5/2}$$

$$\leq \|\nabla f(x^k)\|^2 \left( L_0 + L_1(f(x) - f_\star)^{1/2} - \frac{2}{3}\sqrt{M_k \|\nabla f(x^k)\|} \right).$$

(i) $L_0 \leq L_1(f(x^k) - f_\star)^{1/2}$, then $\quad L_0 + L_1(f(x) - f_\star)^{1/2} - \frac{2}{3}\sqrt{M_k \|\nabla f(x^k)\|}$

$$\leq 2L_1(f(x) - f_\star)^{1/2} - \frac{2}{3}\sqrt{M_k \|\nabla f(x^k)\|} \leq 2L_1\sqrt{D\|\nabla f(x^k)\|} - \frac{2}{3}\sqrt{M_k \|\nabla f(x^k)\|}$$

by inequality (4) with $\alpha = 0.5$ and $M_k = M = \max\left(H, 9L_1^2 D\right)$ yields

$$f\left( x^k - \frac{\nabla f(x^k)}{\sqrt{M \|\nabla f(x^k)\|}} \right) \leq f(x^k) - \frac{1}{3\sqrt{M}} \|\nabla f(x^k)\|^{3/2}.$$

$L_0 \geq L_1(f(x^k) - f_\star)^{1/2}$ reduces to the standard smoothness case with $L = 2L_0$, so that, when

(ii) $L_0 \geq L_1(f(x^k) - f_\star)^{1/2}$ and $(f(x^k) - f_\star)^{1/2} \geq \frac{3}{2}\frac{2L_0 D^2}{\sqrt{HD^3}}$,

then

$$f(x^{k+1}) \leq f(x^k) - \frac{1}{3\sqrt{HD^3}}\left(f(x^k) - f_\star\right)^{3/2}$$

(iii) $L_0 \geq L_1(f(x^k) - f_\star)^{1/2}$ and $(f(x^k) - f_\star)^{1/2} \leq \frac{3}{2}\frac{2L_0 D^2}{\sqrt{HD^3}}$,

then

$$f(x^{k+1}) \leq f(x^k) - \frac{1}{9L_0 D^2}\left(f(x^k) - f_\star\right)^2.$$

Then first two cases reduce to

$$f(x^{k+1}) \leq f(x^k) - \frac{1}{3\sqrt{\max\left(H, 9L_1^2 D\right)D^3}}\left(f(x^k) - f_\star\right)^{3/2},$$

which allows to follow exactly the proof of Theorem E.6 with $H$ replaced with $\max\left(H, 9L_1^2 D\right)$ and $L$ with $2L_0$. $\qquad\square$

## F    MISSING PROOFS FOR CACUSGD

**Theorem F.1** (Theorem 6.3 duplication). *Let Assumption 6.1 holds. Assume that $\widehat{L} \geq 3L$ and $\widehat{H} \geq H$. Then the method (*CaCuSGD*)*

$$x^{k+1} = x^k - g^k / \sqrt{M_k \|g^k\|}, \quad M_k := \begin{cases} \widehat{L}^2 \|g^k\|^{-1}, & \text{if } 4\langle g^k, H^k g^k \rangle^2 \|g^k\|^{-5} \geq \widehat{H}, \\ \widehat{H}, & \text{otherwise,} \end{cases} \quad \text{satisfies}$$

$$\mathbb{E}f(x^{k+1}) \leq \mathbb{E}f(x^k) - \frac{1}{4\widehat{L}}\mathbb{E}\|\nabla f(x^k)\|^2 + \left(\frac{2L}{\widehat{L}^2} + \frac{2L^2}{3\widehat{L}^3}\right)\sigma_g^2 + \frac{\sigma_H^3}{6\widehat{H}^2}, \quad \text{if } \frac{4\langle g^k, H^k g^k \rangle^2}{\|g^k\|^{-5}} \geq \widehat{H},$$

$$\mathbb{E}f(x^{k+1}) \leq \mathbb{E}f(x^k) - \frac{1}{3\sqrt{\widehat{H}}}\mathbb{E}\|\nabla f(x^k)\|^{3/2} + \frac{7}{4\sqrt{\widehat{H}}}\sigma_g^{3/2} + \frac{32}{3\widehat{H}^2}\sigma_H^3, \quad \text{otherwise.}$$

*Proof.* For the sake of simplicity we write $g^k = g(x^k, \xi_k)$ and $H^k = H(x^k, \xi_k)$

There exist $M_k$ is s.t.

$$f_k\left(x^k - \frac{g^k}{\sqrt{M_k \|g^k\|}}\right) \leq f_k(x^k) + \frac{1}{2}\left[\frac{\langle g^k, H^k g^k \rangle}{M_k \|g^k\|} - \frac{1}{\sqrt{M_k}}\|g^k\|^{3/2}\right] - \frac{1}{6\sqrt{M_k}}\|g^k\|^{3/2}$$

$$\leq -\frac{5}{12\sqrt{M_k}}\|g^k\|^{3/2},$$

i.e. $M_k \geq \max\left(H, \frac{4\langle g^k, H^k g^k \rangle^2}{\|g^k\|^5}\right)$. Next we consider the cases.

**Case** $\frac{4\langle g^k, H^k g^k \rangle^2}{\|g^k\|^5} \leq \widehat{H}$. Then $M_k = H$ and

$$-\langle g^k, s^k \rangle + \frac{1}{2}\langle s^k, H^k s^k \rangle + \frac{M_k}{3}\|s^k\|^3 \leq -\frac{5}{12\sqrt{M_k}}\|g^k\|^{3/2},$$

where $s^k = \frac{g^k}{\sqrt{M_k \|g^k\|}}$. Then

$$f(x^{k+1}) \leq f(x^k) - \langle \nabla f(x^k), s^k \rangle + \frac{1}{2}\langle s^k, \nabla^2 f(x^k) s^k \rangle + \frac{H}{3}\|s^k\|^3$$

$$= f(x^k) - \langle \nabla f(x^k) \pm g^k, s^k \rangle + \frac{1}{2}\langle s^k, \nabla^2 f(x^k) s^k \pm H^k s^k \rangle + \frac{H}{3}\|s^k\|^3$$

$$= f(x^k) - \langle g^k, s^k \rangle + \langle s^k, H^k s^k \rangle + \frac{H}{3}\|s^k\|^3$$

$$- \langle \nabla f(x^k) - g^k, s^k \rangle + \frac{1}{2}\frac{\langle g^k, (\nabla^2 f(x^k) - H^k) g^k \rangle}{H\|g^k\|}.$$

Next, one applies Young's inequality ($p = 3, q = 3/2$):

$$uv \leq \frac{1}{3}\alpha^3 u^3 + \frac{2}{3}\alpha^{-3/2}v^{3/2}.$$

with $u = \|s^k\|$, $v = \|\nabla f(x^k) - g^k\|$, $\alpha^3 = H/8$, and

$$-\langle \nabla f(x^k) - g^k, s^k \rangle \leq \frac{1}{24\sqrt{H}}\|g^k\|^{3/2} + \frac{4\sqrt{2}}{3}\frac{1}{\sqrt{H}}\|\nabla f(x^k) - g^k\|^{3/2}.$$

and with $u = \|\nabla^2 f(x^k) - H^k\|$, $v = \frac{\|g^k\|}{2H}$, $\alpha = 2^{5/3}H^{-2/3}$ for

$$\frac{1}{2}\frac{\langle g^k, (\nabla^2 f(x^k) - H^k) g^k \rangle}{H\|g^k\|} \leq \frac{1}{2H}\|\nabla^2 f(x^k) - H^k\|\|g^k\|.$$

The latter gives

$$\frac{1}{2}\frac{\left\langle g^k, (\nabla^2 f(x^k) - H^k)g^k\right\rangle}{H\|g^k\|} \le \frac{1}{24\sqrt{H}}\|g^k\|^{3/2} + \frac{32}{3H^2}\|\nabla^2 f(x^k) - H^k\|^3.$$

Also

$$\|\nabla f(x^k)\|^{3/2} \le \sqrt{2}(\|g^k\|^{3/2} + \|g^k - \nabla f(x^k)\|^{3/2})$$

implying that

$$-\|g^k\|^{3/2} \le -\frac{1}{\sqrt{2}}\|\nabla f(x^k)\|^{3/2} + \|g^k - \nabla f(x^k)\|^{3/2}$$

Putting the pieces together we obtain that

$$f(x^{k+1}) \le f(x^k) - \frac{1}{3\sqrt{H}}\|\nabla f(x^k)\|^{3/2} + \frac{1+4\sqrt{2}}{3\sqrt{2H}}\|g^k - \nabla f(x^k)\|^{3/2} + \frac{32}{3H^2}\|\nabla^2 f(x^k) - H^k\|^3.$$

Taking expectation we have

$$\mathbb{E}f(x^{k+1}) \le \mathbb{E}f(x^k) - \frac{1}{3\sqrt{H}}\mathbb{E}\left[\|\nabla f(x^k)\|^{3/2}\right] + \frac{1+4\sqrt{2}}{3\sqrt{2H}}\sigma_g^{3/2} + \frac{32}{3H^2}\sigma_H^3.$$

**Case** $\frac{4\left\langle g^k, H^k g^k\right\rangle^2}{\|g^k\|^5} > \widehat{H}$. Then $M_k = \frac{\widehat{L}^2}{\|g^k\|}$. We assume that there exists a constatnt $L$ s.t.

$$L_k \stackrel{\text{def}}{=} \frac{\left\langle g^k, H^k g^k\right\rangle}{\|g^k\|^4} \le L,$$

then

$$s^k = \frac{g^k}{\sqrt{M_k\|g^k\|}} = \frac{g^k}{2\widehat{L}}.$$

$$\begin{aligned}
f(x^{k+1}) &\le f(x^k) - \frac{1}{2\widehat{L}}\langle\nabla f(x^k), g^k\rangle + \frac{1}{8\widehat{L}^2}\langle g^k, \nabla^2 f(x^k)g^k\rangle + \frac{H}{24\widehat{L}^3}\|g^k\|^3 \\
&\le f(x^k) - \frac{1}{2\widehat{L}}\langle\nabla f(x^k), g^k\rangle + \frac{1}{8\widehat{L}^2}\langle g^k, (\nabla^2 f(x^k) \pm H^k)g^k\rangle + \frac{H}{24\widehat{L}^3}\|g^k\|^3 \\
&= f(x^k) - \frac{1}{8\widehat{L}}\langle\nabla f(x^k), g^k\rangle + \frac{L_k}{8\widehat{L}^2}\|g^k\|^2 + \frac{H}{24\widehat{L}^3}\|g^k\|^3 \\
&\qquad\qquad\qquad\qquad + \frac{1}{8\widehat{L}^2}\langle g^k, (\nabla^2 f(x^k) - H^k)g^k\rangle
\end{aligned}$$

Next we use that

$$\frac{1}{8\widehat{L}^2}\langle g^k, (\nabla^2 f(x^k) - H^k)g^k\rangle \le \frac{1}{8\widehat{L}^2}\|g^k\|^2\|\nabla^2 f(x^k) - H^k\|,$$

apply Young's ($p = 3, q = 3/2$):

$$ab \le \frac{\alpha^3}{3}a^3 + \frac{2}{3\alpha^{3/2}}b^{3/2}.$$

with $a = \|\nabla^2 f(x^k) - H^k\|$, $b = \frac{\|g^k\|^2}{2\widehat{L}^2}$, $\alpha = 2^{-1/3}H^{-2/3}$ and obtain that

$$\frac{1}{8\widehat{L}^2}\|g^k\|^2\|\nabla^2 f(x^k) - H^k\| \le \frac{1}{6H^2}\|\nabla^2 f(x^k) - H^k\|^3 + \frac{H}{24\widehat{L}^3}\|g^k\|^3.$$

Thus we have

$$f(x^{k+1}) \leq f(x^k) - \frac{1}{2\widehat{L}}\langle \nabla f(x^k), g^k\rangle + \frac{L_k}{\widehat{L}^2}\|g^k\|^2 + \frac{2H}{24\widehat{L}^3}\|g^k\|^3$$
$$+ \frac{1}{6H^2}\|\nabla^2 f(x^k) - H^k\|^3$$
$$\leq f(x^k) - \frac{1}{2\widehat{L}}\langle \nabla f(x^k), g^k\rangle + \frac{L_k}{8\widehat{L}^2}\|g^k\|^2 + \frac{8L_k^2}{24\widehat{L}^3}\|g^k\|^2$$
$$+ \frac{1}{6H^2}\|\nabla^2 f(x^k) - H^k\|^3$$
$$\leq f(x^k) - \frac{1}{2\widehat{L}}\langle \nabla f(x^k), g^k\rangle + \frac{L_k}{4\widehat{L}^2}\|\nabla f(x^k)\|^2 + \frac{2L_k^2}{3\widehat{L}^3}\|\nabla f(x^k)\|^2$$
$$+ \frac{L_k}{4\widehat{L}^2}\|\nabla f(x^k) - g^k\|^2 + \frac{2L_k^2}{3\widehat{L}^3}\|\nabla f(x^k) - g^k\|^2 + \frac{1}{6H^2}\|\nabla^2 f(x^k) - H^k\|^3.$$

Take conditional expectation

$$\mathbb{E}[f(x^{k+1}) \mid x^k] \leq f(x^k) - \Big(\frac{1}{2\widehat{L}} - \frac{L}{4\widehat{L}^2} - \frac{2L^2}{3\widehat{L}^3}\Big)\|\nabla f(x^k)\|^2 + \Big(\frac{L}{4\widehat{L}^2} + \frac{2L^2}{3\widehat{L}^3}\Big)\sigma_g^2 + \frac{\sigma_H^3}{6H^2}.$$

Setting $\widehat{L} > 3L$

$$\mathbb{E}[f(x^{k+1}) \mid x^k] \leq f(x^k) - \frac{1}{4\widehat{L}}\|\nabla f(x^k)\|^2 + \Big(\frac{L}{4\widehat{L}^2} + \frac{2L^2}{3\widehat{L}^3}\Big)\sigma_g^2 + \frac{\sigma_H^3}{6H^2}.$$

And finally, the full expectation

$$\mathbb{E}f(x^{k+1}) \leq \mathbb{E}f(x^k) - \frac{1}{4\widehat{L}}\mathbb{E}\|\nabla f(x^k)\|^2 + \Big(\frac{2L}{\widehat{L}^2} + \frac{2L^2}{3\widehat{L}^3}\Big)\sigma_g^2 + \frac{\sigma_H^3}{6H^2}$$

concludes the proof. $\qquad\square$

## G    EXPERIMENTS

**Logistic regression.** Our first experiment concerns the logistic regression problem with $\ell_2$ regularization:

$$\min_{x \in \mathbb{R}^d} \frac{1}{n} \sum_{i=1}^{n} \left( -b_i \log(\sigma(a_i^\top x)) - (1 - b_i) \log(1 - \sigma(a_i^\top x)) \right) + \frac{\ell}{2} \|x\|^2, \tag{44}$$

where $\sigma \colon \mathbb{R} \to (0, 1)$ is the sigmoid function, $\mathbf{A} = (a_{ij}) \in \mathbb{R}^{n \times d}$ is the matrix of features, and $b_i \in \{0, 1\}$ is the label of the $i$-th sample. We use the 'covtype', 'w8a' and 'mushrooms' datasets, and set $\ell = 0$ or $\ell = 10^{-7}$ to make the problem ill-conditioned. To set $H$ (if applicable), we upper bound the Lipschitz Hessian constant as $\sup_{x \in \mathbb{R}^d} \|\nabla^3 f(x)\| \leq \frac{1}{6\sqrt{3}} \max_i \|a_i\| \|\mathbf{A}\|^2$. This estimate is not tight, which causes non-adaptive methods to converge very slowly.

**Log-sum-exp.** In our second experiment, we consider a significantly more ill-conditioned problem of minimizing

$$\min_{x \in \mathbb{R}^d} \rho \log \left( \sum_{i=1}^{n} \exp \left( \frac{a_i^\top x - b_i}{\rho} \right) \right),$$

where $a_1, \ldots, a_n \in \mathbb{R}^d$ are some vectors and $\rho, b_1, \ldots, b_n$ are scalars. This objectives serves as a smooth approximation of function $\max\{a_1^\top x - b_1, \ldots, a_n^\top x - b_n\}$, with $\rho > 0$ controlling the tightness of approximation. We set $n = 500$, $d = 200$ and randomly generate $a_1, \ldots, a_n$ and $b_1, \ldots, b_n$. After that, we run our experiments for several choices of $\rho$, namely $\rho \in \{0.75, 0.5, 0.25, 0.1, 0.05\}$. The values are specified in brackets on the plots.

Our implementation is available at `https://anonymous.4open.science/r/CaCuTe` and builds on `https://github.com/konstmish/opt_methods`.

### G.1    REGULARIZED NEWTON METHODS (NON ADAPTIVE)

Following (Mishchenko, 2023), we compare CaCuN (Algorithm 1) and AccCaCuN (Algorithm 2) with Cubic Newton (Nesterov & Polyak, 2006), its accelerated modification AccCubic (Nesterov, 2008) and global regularized Newton RegNewton (Mishchenko, 2023) The results are presented in Figures 9 to 12.

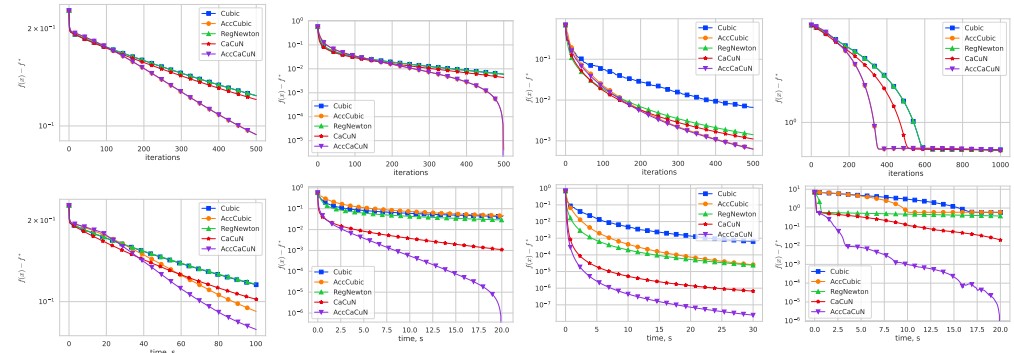

Figure 9: covtype        Figure 10: w8a        Figure 11: mushrooms  Figure 12: LSE (0.05)

The proposed methods mirror the iteration performance of their counterparts but show significantly better wall-clock performance.

### G.2    REGULARIZED NEWTON METHODS (ADAPTIVE)

The adaptive schemes, in contrast, converge after a small number of iterations. Since, cubic Newton with a binary search in regularization is many times slower than AdaN (Mishchenko, 2023) we

compare our adaptive CaCuAdaN and (CaCuAdaN+ algorithms only with AdaN and AdaN+ from (Mishchenko, 2023). The results are presented in Figures 13 to 16.

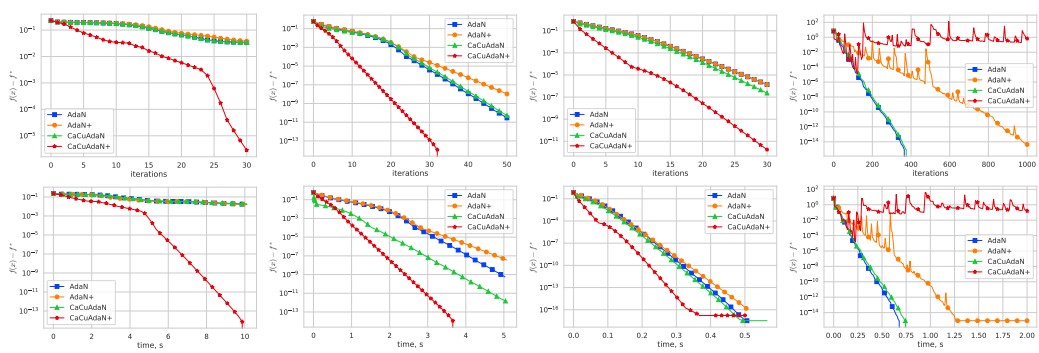

Figure 13: covtype     Figure 14: w8a     Figure 15: mushrooms   Figure 16: LSE (0.05)

Although CaCuAdaN+ significantly outperforms competing methods on logistic regression, it fails to converge on logsumexp due to too optimistic estimates of $H$.

### G.3 FIRST-ORDER METHODS

Following (Malitsky & Mishchenko, 2020), we compare CaCuAdGD (Algorithm 3) with Nesterov's acceleration with Armijo-like line search from (Nesterov, 2013); gradient descent with Polyak stepsize (Polyak, 1987) and AdGD ((Malitsky & Mishchenko, 2020)). The results are presented in Figures 17 to 24.

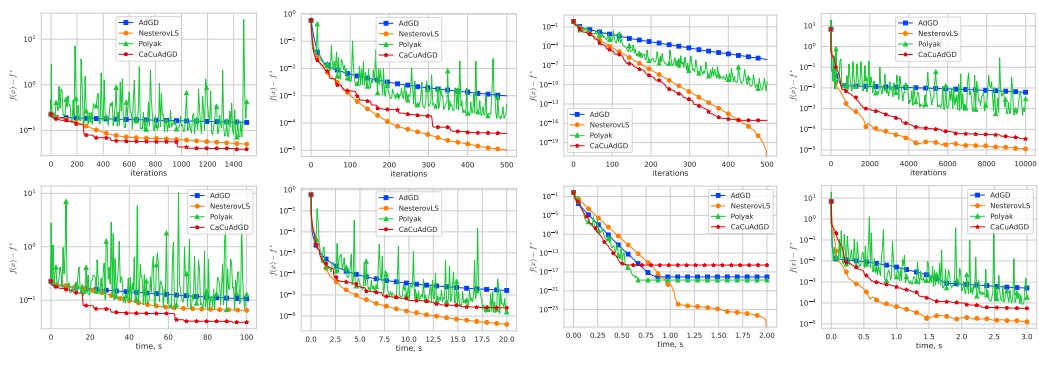

Figure 17: covtype     Figure 18: w8a     Figure 19: mushrooms   Figure 20: LSE (0.05)

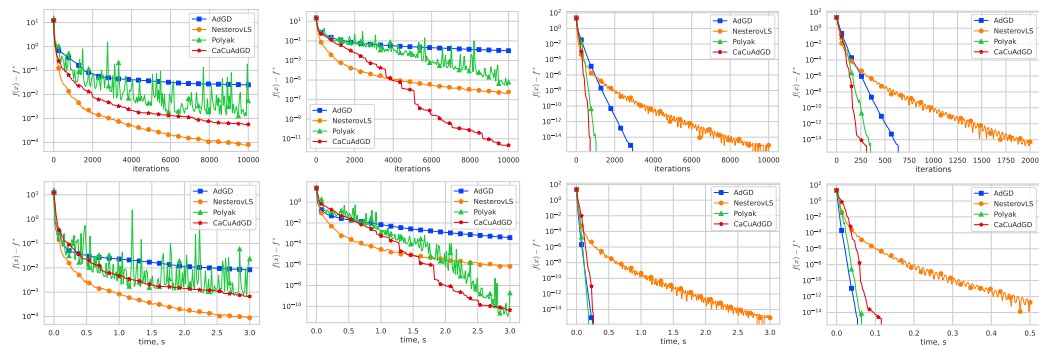

Figure 21: LSE (0.10)    Figure 22: LSE (0.25)    Figure 23: LSE (0.50)    Figure 24: LSE (0.75)

One can see, that CaCuAdGD consistently outperforms AdGD on hard problems and often matches or surpasses Polyak and NesterovLS, otherwise staying competitive.

### G.3.1 FIRST-ORDER METHOD ($\ell_2$ REGULARIZED)

We repeat the experiments with a small $\ell = 10^{-7}$ term in Equation (44) to ensure a unique, well-conditioned optimum and to mirror standard practice. The results are presented in Figures 25 to 28.

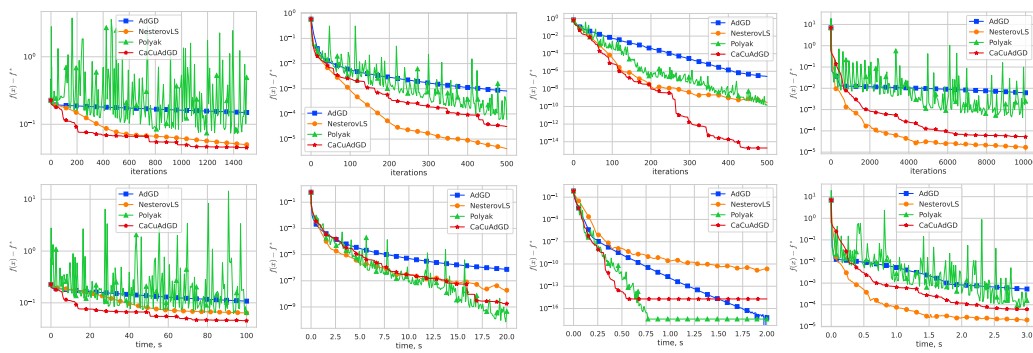

Figure 25: covtype       Figure 26: w8a       Figure 27: mushrooms    Figure 28: LSE (0.05)

One can spot a difference for 'w8a' and 'mushrooms': NewtonLS converges slower, allowing CaCuAdGD to show faster wall-clock performance.

### G.4 STOCHASTIC EXTENSION

In this section we study logistic regression (44) in the stochastic finite-sum setting using SGD ($x^{k+1} = x^k - g^k/\widehat{L}$) and CaCuSGD (Theorem 6.3). We train with shuffled mini-batches of size $n/10$ (drop-last) and sweep $\widehat{L}$ and $\widehat{H}$ (for CaCuSGD). The results are presented in Figures 29 to 31.

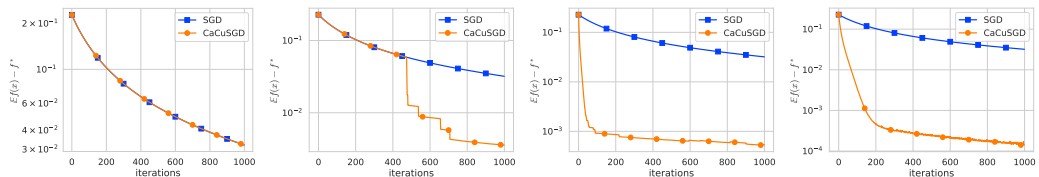

Figure 29: $\widehat{L} = 100$, $\widehat{H} = \{0.1, 1, 10, 100\}$ (from left to right)

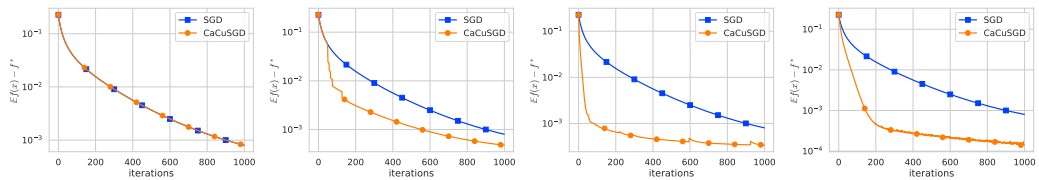

Figure 30: $\widehat{L} = 10$, $\widehat{H} = \{0.1, 1, 10, 100\}$ (from left to right)

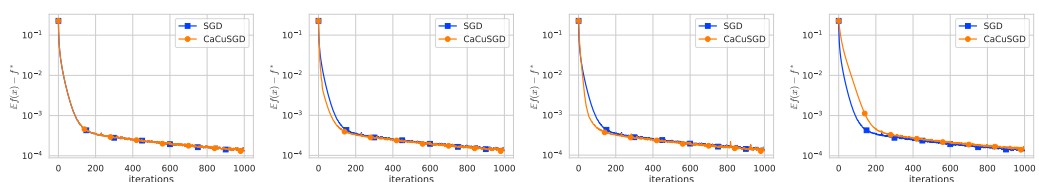

Figure 31: $\widehat{L} = 1$, $\widehat{H} = \{0.1, 1, 10, 100\}$ (from left to right)

Even with a conservative stepsize (large $\widehat{L}$), our scheme rapidly reaches the noise floor. In the rightmost panel ($\widehat{H} = 100$), the floor remains the same but is attained sooner: smaller stepsizes yield a lower floor yet normally slow convergence, whereas CaCuSGD reaches that (smaller) floor faster. Thus one can fix a small stepsize to target a low floor without elaborate tuning, in another words, the method effectively acts as an implicit scheduler.

