# OpenReview forum: "CaCuTe: Casual Cubic-Model Technique for Faster Optimization"
_ICLR.cc/2026/Conference — Submitted to ICLR 2026_

### Official Review · Reviewer_soLp · 2025-10-31

**Soundness:** 2
**Presentation:** 1
**Contribution:** 1
**Rating:** 2
**Confidence:** 4

**Summary:**

This paper considered unconstrained minimization problem $\min f(x)$ where $f$ is twice-differentiable and the Hession of $f$ is $2H$-Liptchitz. The paper studies first-order methods in the form of
$$
x^{k+1} = x^k - \frac{1}{\sqrt{H\|\nabla f(x^k)\|}} \nabla f(x^k).
$$
Combining with cubic regularized Newton steps, the casual cubic Newton method and its acclerated version are proposed. With specific pamameter choices, the casual cubic Newton method recovered the results of the cubic regularized Newton methods. Then under assumptions on bounded Hession eigenvalue in the graident direction, the paper poposed the casual cubic adaptive gradient descent method and extend to directional $L_0, L_1$- smoothness cases. Finally, a variant of methods under stochastic regime is studied.

**Strengths:**

The paper showed a potential framework to analyze cubid regularized Newton methods and gradient methods under various smoothness regimes.

**Weaknesses:**

While the core idea presented in this manuscript is interesting and has potential, the current presentation significantly limits its impact. The contribution of the work is not clearly articulated, and the results are not framed with the necessary rigor to be fully convincing. As a result, the contribution of the work remains unclear; see questions below regrading writing, paper structure and the correctness.

**Questions:**

- $\mathbf I$ is not used in the line 34.


- In Line 59, '...in the convex' seems complete; 'enhances' --> 'is enhanced'?

- In line 69, 'However  ... can be avoided while keeping ... without... under...' this sentence is strange.

- What are the convergence results of the adaptive gradient methods in the related work?


- The sentences and ambiguous words in Section 1.3 are confusing.

    - The first long sentence in Line 89-91 is difficult to understand. I suggest to separate to several sentences. What does the 'corresponds coincides aligns' in the parenthese mean? The terminology 'certificate' is frequently used but I am not sure what method/technique they refer to.

    - In the AccCaCuN paragraph, what does the 'same argument' refer to?

    - What does 'a method is monotone' stand for?

- In section 3,  $O(k^{-2})$ convergence is claimed for Cases I (cubic-model decrease) and $O(k^{-1})$ convergence is claimed for Case II (quadratic-model decrease). Are these claims existing results? If so, proper references should be listed; otherwise, rigorous statements and proofs should be given.

- Why $L$-smoothness is mentioned in Line 186? Is it related to the smoothness assumption in the previous sentence.

- **There is no clear direction linked the proofs in the appendix to the results (theorem, claims, or examples) in the main paper, making it difficult to verify the correctness of the results.**


- Below are questions regarding the results in section 4:

    - Why the inequality in Line 203 holds? It seems inequality (6) is used but $M_k = (3/4)H$ may not satisfied (5).

    - What is $y^0$ in Line 208?

    - What is the gradient step that yields the inequality Line 208? I suggest to write out the first step explicitly.

    -  Section 4 seems to obtain convergence results by relating Algorithm 1 to the results in [Nesterov & Polyak, 2006]. But I am not sure what the 'original proof' refers to and most importantly why Algorithm 1 achieves global $O(k^{-2})$ convergence. The choice of $M_k$ in (5) switches to $M_k = (3/4)H$ and in Algorithm 1 there is no $M_k$. If Algorithm 1 and the proofs are fully covered by the results in [Nesterov & Polyak, 2006]. What is the point of Section 3 since (5) may not hold? Even though the proof is not new, I suggest to write a formal theorem as this is the first main result. The proof can be postponed to appendix.


   - What do the 'same technique' and 'the original converge guarantees' in Section 4.2 refer to?


   - In Algorithm 2, '$M_k \leq H, i.e. M_K = H$' does not make sense.

- The vanishing directional curvature is used in Section 1 and the title of Section 5.1 but never formally introduced. Is it defined as (10)?

- The proof for Theorem B.4 is not self-contained.

- The condition in Theorem 6.3 is inconsistent for the method and the result.

- Can any global convergence rate be derived using the result of Theorem 6.3? How does it compared to other SGD methods?

---

> ### Author Response · Authors · 2025-11-16
>
> We thank the reviewer for the quality and thoroughness; the efforts are truly appreciated. However, we recognize that some parts may not have been understood as intended. For that reason we respectfully disagree with the assertions that our results “are not framed with the necessary rigor” and that “the contribution remains unclear.”
>
> >**While the core idea presented in this manuscript is interesting and has potential, the current presentation significantly limits its impact. The contribution of the work is not clearly articulated, and the results are not framed with the necessary rigor to be fully convincing. As a result, the contribution of the work remains unclear; see questions below regrading writing, paper structure and the correctness.**
>
> The paper clearly itemizes the contributions in Sec. 1.3 (Our Contribution; 6 items), we provide a shortened items below
>
> 1. **Certificate for cubic-model decrease.** We introduce an HVP-based certificate that decides when the gradient step  achieves the cubic-model (CRN) decrease.
> 2. **CaCuN (Casual Cubic Newton).** Using the certificate, CaCuN can skip second-order work while retaining the global $O(k^{-2})$.
> 3. **AccCaCuN (Accelerated CaCuN).** The same idea applies to the accelerated variant, retaining the global $O(k^{-3})$ rate while avoiding second-order work when the certificate holds.
> 4. **CaCuAdGD (Casual Cubic Adaptive GD).** We exploit directional curvature via backtracking (see (14)), requiring only HVPs. When the cubic-model decrease does not apply, the method falls back to the quadratic-model decrease. CaCuAdGD is monotone, parameter-free, and shows strong wall-clock performance compared to \algname{AdGD}.
> 5. **Gradient-directional smoothness relaxation.** We propose an assumption allowing the Hessian eigenvalue in the gradient direction to grow linearly with the square root of the optimality gap. It enables tighter convergence rates and, in particular, a global cubic-model decrease rate.
> 6. **CaCuSGD (Casual Cubic SGD).** Under standard stochastic assumptions with minibatch gradients and a batched Hessian-gradient estimator, our analysis recovers either the stochastic cubic-Newton decrease or the standard stochastic decrease under $L$-smoothness, corresponding to gradient-norm decay $O(1/k)$ or $O(k^{-1.5})$.
>
> Formal assumptions appear in Sec. 2 (Assumptions 2.1 to 2.3), Sec. 5 (Assumptions 5.5 and 5.11; note that the directional smoothness in Sec. 5.1 is illustrative and intentionally not formalized as an assumption), and Sec. 6 (Assumption 6.1). The main results are Theorem 4.1, Theorem 5.8, Corollary 5.15, and Theorem 6.3, with complete, self-contained proofs in Appendices B, D, and E. We appreciate the observation that Sec. 4.1 (Casual Cubic Newton) presents counterparts for core inequalities used in the original proof, which may not make the result immediately obvious; despite the brevity, the result is rigorously proved. Summing up, we could not identify any specific theorem, assumption, or step that lacks rigor. We invite the reviewer to point to any concrete issues so that we can address them precisely.
> To aid readability, we will (a) add explicit cross-references from each contribution and theorem to its proof in the appendix, and (b) provide a standalone theorem and proof for the result in Sec. 4.1 (Casual Cubic Newton).
> Given the above, we believe the critique is not substantiated and ask the reviewer to reconsider the reject recommendation unless concrete issues are identified. Below we address the reviewer’s questions. We are also grateful for the other minor issues and typos identified.
>
>
>
> >**What are the convergence results of the adaptive gradient methods in the related work?**
> To the best of our knowledge all the non-accelerated methods provide only global $O(k^{-1})$ rate. We believe that our results are the first one providing (a) local $O(k^{-2})$ rate, (b) global $O(k^{-2})$ for the relaxed smoothness (when L_0 = 0). For future works we plan to accelerate CaCuAdGD and obtain local $O(k^{-3})$.
>
> >**The sentences and ambiguous words in Section 1.3 are confusing.**
> **The first long sentence in Line 89-91 is difficult to understand. I suggest to separate to several sentences. What does the 'corresponds coincides aligns' in the parenthese mean?**
> We missed to proofread this sentence. The right version should have been as follows:
> We introduce a Hessian-vector-product-based certificate that determines when the decrease from the gradient step $x^{k+1} = x^k - {\nabla f(x^k)}/{\sqrt{H_k\|\nabla f(x^k)\|}}$ matches the decrease guaranteed by the cubic-regularized Newton (CRN) step, where $H_k$ is a local estimate of the Hessian-Lipschitz constant

---

> ### Author Response · Authors · 2025-11-16
>
> >**The terminology 'certificate' is frequently used but I am not sure what method/technique they refer to.**
> The proposed technique establishes the certificate - the descent inequality (6), which is an implication of condition (5). In Line 192 we write that (6) (or (5), since (5) implies (6)) certificates for skipping  the second order work.
>
> >**In the AccCaCuN paragraph, what does the 'same argument' refer to?**
> The same argument refers to the item right above (Line 094), i.e. the certificate enables provably obviating the second order work.
>
> >**What does 'a method is monotone' stand for?**
> The objective value monotonically decreases along the trajectory. We indeed realized that the terminology may be not widely used, we will define it more precisely.
>
> >**In section 3, $O(k^{-2})$ convergence is claimed for Cases I (cubic-model decrease) and $O(k^{-1})$ convergence is claimed for Case II (quadratic-model decrease). Are these claims existing results? If so, proper references should be listed; otherwise, rigorous statements and proofs should be given.**
> Section 3 provides the descent inequalities (7), (8), the rigorous statements and proofs are in the appendix. Ones the inequalities obtained, the claimed rates simply follow then from convexity (see Lines 1068-1082). The results are formalized in Lemma D.4.+Corollary D.5 with the proper reference (Line 1129 explicitly refers to Nesterov, 2022, Lemma A.1) and proofs. Although it may not be obvious, claimed rates in Section 3 provided as a motivation, the exact applications for obtaining rates of these inequalities vary for different proofs.
>
> >**Why $L$-smoothness is mentioned in Line 186? Is it related to the smoothness assumption in the previous sentence.**
> The previous sentence is indefinite about smoothness – **an** additional smoothness. The constant $L$ appears in our result, despite the fact that we do not assume $L$-smoothness. It follows from Assumptions 2.1, 2.2 (Line 187). Also we introduce the relaxation in Assumption 5.11, which allows global $O(k^{-2})$ rate (when L_0 = 0).
>
> >**Why the inequality in Line 203 holds? It seems inequality (6) is used but $M_k = (3/4)H$ may not satisfied (5).**
> The inequality in Line 203 may not hold, it is being checked instead. If it holds, then the algorithm obviates the second order work. Indeed, $M_k = (3/4)H$ may not satisfy (5), then the inequality in Line 203 may not hold, and cubic Newton step should be done.
>
> >**What is $y^0$ in Line 208?**
> Thanks, that is a typo, $x^0$ should be here.
>
> >**Section 4 seems to obtain convergence results by relating Algorithm 1 to the results in [Nesterov & Polyak, 2006]. But I am not sure what the 'original proof' refers to and most importantly why Algorithm 1 achieves global $O(k^{-2})$ convergence. … Even though the proof is not new, I suggest to write a formal theorem as this is the first main result. The proof can be postponed to appendix.**
> We indeed realized that it may not be obvious, and it was better to formalize the result as a self contained theorem. The rate is proven in Theorem 4 from [Nesterov & Polyak, 2006]. Inequlity (9) + convexity implies inequlity in Lines 1079 (see Lines 1074-1079 for the derivation), which is exactly the inequality appearing in [Nesterov & Polyak, 2006]. So it holds in both cases, and one can follow the rest of the proof in Theorem 4 from [Nesterov & Polyak, 2006] to obtain global $O(k^{-2})$. It remains to follow the original proof in the other case for the first iterate, which requires the inequality in Lines 207-209. We will definitely introduce a separate theorem with the self-contained proof.

---

> ### Author Response · Authors · 2025-11-16
>
> >**What is the point of Section 3 since (5) may not hold?**
> To be precise let us say “M_k is not chosen according to (5)” instead of “(5) does not hold”. Inequality (5) claims the existence of $M_k$ s.t. a descent inequality holds  and states the value explicitly under Assumption 2.1. If M_k is chosen according to (5), then one of the descent inequalities (7) or (8) hold  – the point of Section 3. But (5) is not a criterion, meaning (5) implies (7) or (8), but (7) or (8) may hold even when M_k is not chosen according to (5).
>
> >**The choice of $M_k$ in (5) switches to $M_k = (3/4)H$ and in Algorithm 1 there is no $M_k$. … **
> The choice $M_k = (3/4)H$ refers to Inequality (6); Inequality (9) is the result of the substitution $M_k = (3/4)H$ into (6). Note, there is no need to use (5), we check only its implication (6). Even if M_k is not chosen as (5) suggests, (6) may hold. If (6) holds, the algorithm skips the second order work. If it does not, the algorithm falls back to cubic Newton.
>
> >**If Algorithm 1 and the proofs are fully covered by the results in [Nesterov & Polyak, 2006].**
> [Nesterov & Polyak, 2006] covers the only case, which uses computationally expensive second order information. We extend the analysis to the other case - the gradient step.
>
> >**What do the 'same technique' and 'the original converge guarantees' in Section 4.2 refer to?**
> It refers to the technique in the subsection above, which is checking Inequality (6) with $M_k = (3/4)H$, and replacing the expensive cubic Newton step with the gradient step, while preserving global $O(k^{-3})$.
>
> >**In Algorithm 2, '$M_k \leq H, i.e. M_K = H$' does not make sense.**
> $H$ is a global bound on the Hessian smoothness, it holds for any point according to Assumption 2.1. However, for some points the local value $H_k$ may be much smaller. But, the derivation (Lines 141-155) of the descent Inequality (6) requires only some $M_k > H_k$, meaning that $M_k$ may be less than the global value $H$, i.e $M_k \leq H$ . The conservative choice $M_k = H$ is always good in terms of theory, but may lead to smaller stepsizes. Further, Inequality (6) may hold even when $M_k < H_k$, but it is not theoretically guaranteed. Even if Inequality (6) does not hold, one always has the fall back option -- cubic Newton step with $H$, so any value of $M_k$ can be set in Algorithm 2. But small optimistic values may never satisfy Inequality (6); in this case our method reduces to Aceelerated CRN.
>
> >**The vanishing directional curvature is used in Section 1 and the title of Section 5.1 but never formally introduced. Is it defined as (10)?**
> Yes; it is introduced in (10). Section **5.1 Vanishing Directional Curvature** should make make this explicit. Moreover, we intentionally do not wrap (10) as Assumption -- the assumption is non-standard and serves mostly for illustrative purposes since it allows us to simply obtain a global $O(k^{-2})$ rate. But it still motivates us to consider a much more interesting relaxation of the standard smoothness -- Assumption 5.11, which covers the standard L-smoothness and allows to state the global $O(k^{-2})$ rate if $L_0=0$. This assumption is more realistic since it holds  for separable logistic regression loss (Example 5.14).
>
> >**The proof for Theorem B.4 is not self-contained.**
> Thanks, we indeed missed to show a division of both sides to A_k and refer to the result right above stating that A_k grows as $O(k^3)$. But we believe that the proof cannot be considered not self-contained for that reason.
>
> >**The condition in Theorem 6.3 is inconsistent for the method and the result.**
> Thanks for spotting a typo! The power should be 5 instead of -5 in Line 440.
>
> >**Can any global convergence rate be derived using the result of Theorem 6.3? How does it compared to other SGD methods?**
> As discussed below the theorem, each of the resulting inequality solely recovers either convergence of SGD, or stochastic cubic Newton in the nonconvex setting – gradient norm vanishes either as 1/k or 1/k^1.5. But combining these two bounds seems challenging. However, the authors believe that a result is possible in the convex setting.

---

> > ### Comment · Reviewer_soLp · 2025-11-25
> >
> > Thank the authors for their response.
> >
> > > Section 3 provides the descent inequalities (7), (8), the rigorous statements and proofs are in the appendix. ....
> >
> > > We indeed realized that it may not be obvious, and it was better to formalize the result as a self contained theorem.
> >
> > At the moment, there is no appendix containing the proofs for CaCuN. I appreciate the explanation given in the response, but these arguments need to be explicitly stated and included in the paper.
> >
> > Currently, Sections 3 and 4 mix together assumptions from prior work, reused techniques, new assumptions, new arguments, and the proposed certificate-based criterion. For clarity, the paper needs to separate what is inherited from existing theory and what is novel, for example (but not necessarily) through subsections such as:
> >
> >  - Previous assumptions and known results
> >  - New assumptions / new structural conditions
> >  - New inequalities and their proofs
> >  - How the certificate uses these conditions to skip the second-order work.
> >
> > Without this restructuring, it is unreasonable to expect readers to verify the correctness of the results, particularly when many nontrivial arguments appear only in the response and not in the paper.
> >
> > > Note, there is no need to use (5), we check only its implication (6). Even if M_k is not chosen as (5) suggests, (6) may hold. If (6) holds, the algorithm skips the second order work. If it does not, the algorithm falls back to cubic Newton.
> >
> > If (5) is not used and the algorithm only checks (6), then it is unclear why (5) is needed at all. A similar issue applies to (7) and (8): these inequalities are not referenced or used in Algorithm 1.
> >
> > > The conservative choice $M_k = H$ is always good in terms of theory, but may lead to smaller step sizes.
> >
> > If I understand correctly, $M_k = H$ in Algorithm 2. Then $M_k \leq H$ should be removed as this is not equivalent to $M_k = H$.
> >
> > The results in Section 5 and Section 6 are appealing but the motivation and theoretical foundation—contained in Sections 3 and 4—requires substantial rewriting and clarification before the contributions can be properly evaluated.

---

> > > ### Author Response · Authors · 2025-11-25
> > >
> > > >>Section 3 provides the descent inequalities (7), (8), the rigorous statements and proofs are in the appendix. ....
> > > >
> > > >>We indeed realized that it may not be obvious, and it was better to formalize the result as a self contained theorem.
> > > >**At the moment, there is no appendix containing the proofs for CaCuN. I appreciate the explanation given in the response, but these arguments need to be explicitly stated and included in the paper.**
> > > >
> > > >As noted above, we will include a standalone CaCuN theorem and a full appendix proof. If anything still seems unclear, could you please suggest the specific edits?
> > >
> > > >**Currently, Sections 3 and 4 mix together assumptions from prior work, reused techniques, new assumptions, new arguments, and the proposed certificate-based criterion.**
> > > >
> > > > In Section 3 we use only the standard Hessian–Lipschitz assumption (Assumption 2.1). Right before Assumption 2.1 we clearly cite Nesterov & Polyak (2006) and also state the very standard convexity assumption (Assumption 2.2).
> > > >We additionally use bounded level sets (Assumption 2.3), which is standard and appears, for example, in Nesterov (2008).
> > > >Moreover, we explicitly cite all reused techniques at the point of use.
> > > >We respectfully disagree with the claim of mixing. If you believe specific places are conflated, please indicate the exact lines/equations, and we will address them directly.
> > >
> > >
> > > >**For clarity, the paper needs to separate what is inherited from existing theory and what is novel, for example (but not necessarily) through subsections such as:**
> > > >
> > > >**1. Previous assumptions**
> > > >
> > > > We only use standard Assumptions 2.1, 2.2, 2.3.
> > > >
> > > > **...and known results**
> > > >
> > > > In Section 1.3 we state that we retain the known convergence rates, but skip the second-order work when possible.
> > > >
> > > >**New assumptions / new structural conditions**
> > > >
> > > > Section 1.3 states our contribution **Gradient-Directional Smoothness Relaxation**, which is the only new assumption relaxing standard $L$-smoothness.
> > > >
> > > >**New inequalities and their proofs**
> > > >
> > > > Descent inequalities with the proof are in Section 3. Note, we do not list the new descent inequalities as a standalone contribution; they constitute the certificate mechanism.
> > > >
> > > >**How the certificate uses these conditions to skip the second-order work.**
> > > >
> > > > Could the reviewer specify the “conditions”? To obtain the certificate we only use the standard **Hessian–Lipschitz** assumption. One either chooses $M_k$ as (5) suggests ( which imply (6)), or directly picks $M_k$ that satisfies (6). For CRN type methods one can check (6) with any $M_k \leq H_k$, for CaCuAdGD we use (5).
> > > >
> > > >
> > > > To the best of our understanding, the main concern is our contribution and its relation to known results. Our main observation is that second-order work is sometimes unnecessary and can be replaced by a first-order step. **Hessian–Lipschitz** and **convexity** are the standard assumptions to analyze CRN-type methods. Accelerated CRN also requires Assumption 2.3. Our analysis is under these assumptions.
> > > > Nevertheless, we see the reviewer’s point that a clear separation of what is inherited and what is novel is useful. However, we believe the paper already contains the necessary citations and remarks for reused techniques **and does not claim these techniques as contributions**. Our contributions summarize as follows
> > > >
> > > > - **Certificate for Cubic-Model Decrease.** When the gradient-step decrease corresponds to the cubic-model (i.e., CRN) step decrease.
> > > > - **CaCuN.** The certificate enables provably obviating second-order work while retaining global $O(k^{-2})$.
> > > > - **AccCaCuN.** Retains global $O(k^{-3})$ while skipping second-order work when certified.
> > > > - **CaCuAdGD.** The central contribution addressing infeasibility of direct second-order methods. **We use (5)** to enable CRN-type decrease when possible.
> > > > - **Gradient-Directional Smoothness Relaxation.** Allows the Hessian eigenvalue in the gradient direction to grow linearly with the square root of the optimality gap; relaxes standard $L$-smoothness and enables tighter rates.
> > > > - **CaCuSGD.** Under standard assumptions with minibatch gradients and a batched HVP estimator, our analysis recovers either the stochastic cubic-Newton decrease or the standard stochastic decrease.

---

> > > ### Author Response · Authors · 2025-11-25
> > >
> > > >>Note, there is no need to use (5), we check only its implication (6). Even if M_k is not chosen as (5) suggests, (6) may hold. If (6) holds, the algorithm skips the second order work. If it does not, the algorithm falls back to cubic Newton.
> > > >
> > > >**If (5) is not used and the algorithm only checks (6), then it is unclear why (5) is needed at all. A similar issue applies to (7) and (8): these inequalities are not referenced or used in Algorithm 1.**
> > > >Please note that the initial concern was about Algorithm 1 (CaCuN). Indeed, Algorithm 1 uses neither (5) nor (8). The goal for Algorithm 1 is keeping the $O(k^{-2})$ rate via (7). Only one option in (5) implies (7). So (7) can be checked directly. That would lead to a different constant compared to the original bound from Nesterov & Polyak (2006). But we wanted to preserve the constant. To do this we check a more strict version of (7) appearing in Algorithm 1 Line 3.
> > > > Beyond Algorithm 1, there is Algorithm 3. It uses (5) implicitly. The expression for $M_k$ given by (5) is substituted to (6), which is exactly Line 6 in Algorithm 3.
> > >
> > > >>The conservative choice $M_k = H$ is always good in terms of theory, but may lead to smaller step sizes.
> > > >
> > > >**If I understand correctly, $M_k = H$ in Algorithm 2. Then $M_k \leq H$ should be removed as this is not equivalent to $M_k = H$.**
> > > >
> > > >The analysis holds for any choice $M_k \leq H$. Using a smaller M_k only makes the check in Line 11 of Algorithm 3 stricter, so the guarantees of Theorem 4.1 are preserved (you may skip the second-order step less often, but never incorrectly).
> > > >
> > > >If $M_k \leq H$ is confusing, we can set $M_k = H$ in the pseudocode and add a remark: “Any $M_k \leq H$ yields the same correctness guarantees; smaller M_k only tightens the certificate and may trigger the CRN fallback more often.” We’re happy to make this change if it helps readability.
> > >
> > >
> > > >**Without this restructuring, it is unreasonable to expect readers to verify the correctness of the results, particularly when many nontrivial arguments appear only in the response and not in the paper.**
> > > >
> > > >**The results in Section 5 and Section 6 are appealing but the motivation and theoretical foundation—contained in Sections 3 and 4—requires substantial rewriting and clarification before the contributions can be properly evaluated.**
> > > >
> > > >We kindly but strongly disagree that the contributions cannot be properly evaluated as written. The core inequalities are presented and proved; for CaCuN we explain how to modify the original proof; the other proofs are in the appendices; the algorithms are explicitly stated and the conditions being checked are spelled out. However, we acknowledge that some parts are not easy to verify at a glance. In the revision we will:
> > > >
> > > >- Comment on prior assumptions/results vs. ours, and separate them clearly.
> > > >- Support our use of standard assumptions with explicit citations.
> > > >- Add a **standalone CaCuN theorem** with a complete proof.
> > > >- Clarify the concerns of $H_k$ and $M_k$, and define the certificate precisely as (6) (in a form valid for any $M_k \le H_k$). Algorithms 1–2 will **explicitly check (6)** (with $M_k=\tfrac{3}{4}H$) and fall back to CRN if it fails.
> > > >- Clarify that **Eq. (10) is illustrative assumption** and that the central formal relaxation is **Assumption 5.11**.
> > > >- Emphasize **Assumption 5.11** as a main contribution.
> > > > -add a concise remark on how the certificate skips second-order work under.
> > > >- Patch the other minor gaps (e.g., the \(A_k\) step, “same argument” wording, etc.).
> > >
> > > Given the tight timeline, please confirm **as soon as possible** that these changes address your concerns, or indicate any specific lines/equations we should adjust further.

---

### Official Review · Reviewer_w1tX · 2025-10-31

**Soundness:** 3
**Presentation:** 2
**Contribution:** 3
**Rating:** 6
**Confidence:** 4

**Summary:**

This paper presents **CaCuTe**, a family of algorithms that leverage **cubic regularization ideas** while avoiding the heavy cost of full cubic‐model solves.
Starting from the **Lipschitz–Hessian (2H–smooth)** assumption used in classical cubic‐regularized Newton (CRN) methods, the authors derive a **Hessian–vector–based condition** that certifies when a **simple gradient step** will achieve the same decrease as a cubic step.
This leads to the **CaCuN algorithm**, which dynamically switches between a cheap gradient step and a standard CRN step depending on the local curvature.
The approach extends to an **accelerated variant (AccCaCuN)** that retains the accelerated \(O(k^{-3})\) global rate, as well as a **first-order adaptive method (CaCuAdGD)** that uses Hessian–vector products and backtracking to adapt \(M_k\) without explicit Hessians.
A **stochastic version (CaCuSGD)** further generalizes the idea to noisy gradients.
Overall, the paper provides a unified view of cubic-model techniques that reduce computational cost while maintaining theoretical guarantees.

**Strengths:**

- **Clear motivation:** Classical CRN guarantees are strong but expensive; CaCuTe identifies when cheaper first-order steps suffice.
- **Theoretical soundness:** The paper rigorously derives one-step decrease bounds, global \(O(k^{-2})\) and accelerated \(O(k^{-3})\) rates, and expected stochastic analogues.
- **Low overhead:** The curvature certificate requires only **one Hessian–vector product**, a modest cost compared to full second-order solves.
- **Comprehensive framework:** Covers deterministic, accelerated, adaptive, and stochastic regimes under a single analytic template.
- **Practical relevance:** Algorithms can exploit automatic-differentiation HVPs available in modern ML libraries.

**Weaknesses:**

- **Complex exposition:** The notation is dense and the main ideas are sometimes buried in algebraic detail; a higher-level intuition for why the curvature test works would help readers.
- **Dependence on convexity:** The theoretical results assume convex objectives; it is unclear how the approach behaves on **non-convex** losses where Hessians may be indefinite.
- **Parameter sensitivity:** The constants used in the curvature condition (e.g., the choice of \(H\) and \(\alpha\)) could affect practical performance, but this dependence is not discussed.
- **Weak CaCuSGD results:** The empirical performance of **CaCuSGD** appears weak compared to standard **SGD**. It would be interesting to see whether adding **momentum or variance reduction** could mitigate stochastic noise.
- **Reliance on differing assumptions:** The analysis invokes different assumptions (e.g., Eq. (10), Eq. (14)), making it difficult to compare results across variants. A **summary table** of assumptions and corresponding guarantees would greatly clarify the contributions.
- **Quantification missing:** The paper does not quantify **how often the full cubic Newton step is avoided** in practice. Reporting this ratio would concretely demonstrate the computational advantage claimed.

**Questions:**

1. Could the same idea extend to **non-convex** settings?
2. In practice, how often does CaCuN actually switch to the cubic step versus taking the cheaper gradient step?
3. What is the **computational overhead** of the backtracking procedure in CaCuAdGD relative to standard gradient descent?

---

> ### Author Response · Authors · 2025-11-16
>
> Thank you for the positive evaluation of our work. Several of the listed weaknesses are either already addressed in the paper or stem from visibility/presentation rather than missing content. A few items are matters of emphasis rather than substantive gaps. With these clarifications in place, we respectfully ask the reviewer to reconsider and raise the overall score.
>
> >**Complex exposition**
> Our analysis relies on two simple descent inequalities derived in Section 3, in Line 177 we state that one of them corresponds to an $O(k^{-2})$ rate. The higher level intuition is summarized in Section 5.4. Please let us know if further clarification would be helpful.
>
> >**Dependence on convexity**
> Thanks for the question! That is indeed a very interesting direction to study, we partially did it for the stochastic setting -- we do not assume convexity. However, in Lines 444-445 we state that telescoping each of these one-step descent inequalities is a challenging task. The same holds also for the deterministic setup. More broadly, convexity often guides practical algorithm design. Recent adaptive methods like D-Adaptation [1] and Prodigy [2] show how convexity-inspired ideas enhance nonconvex optimization.
>
> >**Parameter sensitivity**
> Indeed, the choice of $H$ may affect the performance. We experiment with the theoretical value proposed in (Mishchenko, 2023). We also propose adaptive methods and compare them against competitors in Sections C and F.2, which do not require a global value H and adapt to the local curvature. Next, we provide extensive discussion on the choice of alfa in Section 5.4. Please let us know if further clarification would be helpful.
>
> >**Weak CaCuSGD results**
> We respectfully disagree with the comment. Our stochastic variant triggers curvature steps only when the test certifies a larger guaranteed decrease; otherwise it falls back to SGD. As discussed below the theorem, each of the resulting inequality solely recovers either convergence of SGD, or stochastic cubic Newton in the nonconvex setting -- gradient norm vanishes either as 1/k or 1/k^1.5. Unfortunately, combining these two bounds seems challenging, but one may expect something in between in practice.
> Momentum and variance reduction are interesting directions, but they are outside the main scope of CaCuTu; a thorough investigation  would require separate analysis and is deferred to future work.
>
> >**Reliance on differing assumptions**
> Note, that equation (10) intentionally is not wrapped as Assumption – the assumption is non-standard and serves mostly for illustrative purposes since it allows us to simply obtain a global $O(k^{-2})$ rate. But it still motivates us to consider a much more interesting relaxation of the standard smoothness – Assumption 5.11, which covers the standard L-smoothness and allows to state the global $O(k^{-2})$ rate if $L_0=0$. This assumption is realistic since it holds for separable logistic regression loss (Example 5.14).
>
> >**Quantification missing**
> Unfortunately, we do not have any theoretical result on “expected” oracle costs. However we can comment on experimental behaviour. CaCuN type methods are good in non adaptive settings, i.e. global $H$ is known. But with the adaptive methods available (see Appendix C), proposed methods do not require second order at first iterates at all. For the further iterations, we noticed that local value $H_k$ considerably changes -- after a successful (high objective value decrease) the $H_k$ dramatically increases. After that a few gradient steps are done to recover let’s say.
>
> [1] Defazio, Aaron, and Konstantin Mishchenko. "Learning-rate-free learning by d-adaptation." International Conference on Machine Learning. PMLR, 2023.
>
> [2] Mishchenko, Konstantin, and Aaron Defazio. "Prodigy: An expeditiously adaptive parameter-free learner." arXiv preprint arXiv:2306.06101 (2023).

---

### Official Review · Reviewer_LXQv · 2025-11-01

**Soundness:** 2
**Presentation:** 2
**Contribution:** 1
**Rating:** 2
**Confidence:** 4

**Summary:**

**Summary**

This paper establishes the convergence rate of a special stepsize rule for gradient descent under different settings. The stepsize rule is motivated by the optimality condition of the subproblem arising from the cubic regularized Newton's method. Some experiments validate the efficiency of the proposed method.

**Strengths:**

**Strength**

The paper is overall easy to follow.

**Weaknesses:**

**Weaknesses**

I have several concerns regarding both the theoretical and practical aspects of the paper.

1. Theoretical complexity

   In terms of the theoretical complexity, while the paper claims $O(1/K^2)$ result is achievable using first-order information, the analysis ends up relying on strong assumptions such as equation (10). In general, the first-order methods developed in the paper only guarantee an $O(1/K)$ convergence rate. I find the result relatively weak since it uses extra operations such as the Hessian vector product. Besides, some results (**Corollary 5.7**) in the paper contain strong assumptions on the trajectory, which I find unacceptable; some results (**Theorem 6.3**) are incomplete and do not give a convergence rate.

2. Practical implementation

   The proposed algorithm is claimed to be parameter-free. However, in most cases, it still requires knowledge of the Hessian Lipschitz constant $H$. Even the adaptive variant still requires a line-search procedure and introduces an additional hyperparameter $\alpha$. Given that the algorithm essentially adopts the gradient direction, I don't think the algorithm design is justified.

3. Weak experiment evaluation

   The experiments in **Section 7**mseem cherry-picked. The results from the appendix suggest the proposed stepsize typically underperforms standard accelerated gradient descent.

Finally, I noticed a number of typos and notation inconsistencies throughout the paper. Overall, I don't think the paper can be published at ICLR in its current form.

**Questions:**

**Questions**

1. I don't feel I understand what "casual" in the title means. Does it come from the motivation on line 41?
2. Except for (AccCaCuN), the algorithms developed in the paper search along the gradient direction. Is there any intuition that the resulting algorithm can outperform vanilla gradient descent?

**Minor issues**

1. Line 11

   You mention "local" rate here, but the contributions claim global convergence.

2. Line 93

   "i.e. logistic regression" is unclear.

3. Line 130

   "is convex with, i.e." is unclear.

4. Line 132

   I don't think continuity is necessary here.

5. Line 160

   "Inequality (5) employs..." is unclear.

6. Line 186, 254, 284

   Gradient smoothness corresponds to Lipschitz Hessian; Hessian $\nabla^2 f$  corresponds to the Lipschitzness of the third-order derivative.

7. Line 356

   The assumption is not correct

**Details Of Ethics Concerns:**

N/A.

---

> ### Author Response · Authors · 2025-11-16
>
> We thank the reviewer for their time. However, most objections are unsupported by specific evidence or contradict the paper’s content. Several points mix sufficient conditions used for illustration with the standing assumptions of our main results; others overlook explicit discussions (e.g., beneath Theorem 6.3 or Line 186) and ignore comparison with concurrent methods.
> Crucially, we were unable to identify a single incorrect theorem, missing step, or violated assumption. We therefore believe the low scores and the reject recommendation are not warranted and ask the reviewer to reconsider in view of the detailed point-by-point responses below.

---

> ### Author Response · Authors · 2025-11-16
>
> >**In terms of the theoretical complexity, while the paper claims $O(1/K^2)$ result is achievable using first-order information, the analysis ends up relying on strong assumptions such as equation (10).**
> Note that Equation (10) is intentionally not stated as an assumption; it is non-standard and serves mostly for illustrative purposes since it allows us to obtain the global $O(k^{-2})$ rate. However, it still motivates us to consider a much more interesting relaxation of the standard smoothness, Assumption 5.11, which covers the standard $L$-smoothness and allows us to establish the global $O(k^{-2})$ rate if $L_0=0$. This assumption is realistic since it holds for the separable logistic regression loss (Example 5.14).
>
>
>
>
>
>
> >**In general, the first-order methods developed in the paper only guarantee an $O(1/K)$ convergence rate. I find the result relatively weak since it uses extra operations such as the Hessian vector product.**
> That said, the case when $L_0=0$ occurs under a slightly different $L_0,L_1$ smoothness relaxation in the cutting-edge application [1]. In this case, a global $O(k^{-2})$ rate is proved by Corollary 5.15. Therefore, we view this as a significant direction for further study. Moreover, Hessian vector products (HVPs) are readily supported in modern frameworks (e.g., PyTorch) yet remain underutilized; with our experiments we show that exploring them has substantial potential. Further, CaCuAdGD is a non-accelerated method and it cannot be better than $O(k^{-1})$. Although accelerated methods enable faster rates, this should not stop studying non-accelerated methods. Moreover, while accelerated methods can achieve faster worst-case rates, this does not diminish the relevance of non-accelerated methods.For example, non-accelerated methods achieve linear convergence under strong convexity or the PL condition even without knowing the parameter, and they converge faster near minimisers. IKnown accelerated schemes  require an estimate of the strong-convexity parameter to guarantee linear convergence, whereas gradient descent does not. Our experiments also show that on well-conditioned problems accelerated variants can be slower (see Figures 23 and 24). Beyond rates, our contribution is conceptual: we identify a realistic regime – Assumption 5.11 with $L_0=0$ – under which a global $O(k^{-2})$ rate is provable, and we show that Hessian-vector products, which are readily supported in modern frameworks, can improve performance but remain underused. For these reasons, we do not view a non-accelerated $O(k^{-1})$ rate as a weak result; rather, it is a meaningful step that opens a promising line of research.
> We would appreciate the opportunity to discuss these points further and address any remaining concerns.
>
> >**Line 186, 254, 284 Gradient smoothness corresponds to Lipschitz Hessian; Hessian corresponds to the Lipschitzness of the third-order derivative.**
> We believe there is a shift in terminology. Gradient $L$-smoothness means that the gradient is $L$-Lipschitz, which for twice-differentiable $f$ is implied by $\|\nabla^2 f(x)\|\le L$ on the relevant set. By contrast, “Lipschitz Hessian” is a stronger, third-order condition.
> In Lines 186-189 we state that the standard smoothness (gradient $L$-smoothness) holds under Assumption 2.3 because the methods are monotone: the trajectory stays in the compact sublevel set $\mathcal L_0$, and since $f$ is twice differentiable, $L=\sup_{x\in\mathcal L_0}\|\nabla^2 f(x)\|<\infty$. This shows that a finite $L$ exists, even if it may be large. However, inequality (10) does not hold in general even on a compact set, so Line 254 is needed for the sake of rigor.
> Assumption 5.11 (Line 356) relaxes standard smoothness: condition (14) holds with $L_0=\sup_{x\in\mathcal L_0}\|\nabla^2 f(x)\|$ and $L_1=0$, so the standard case is covered. See also Remark 5.12, which follows from the Cauchy-Schwarz inequality:
> $ \frac{\langle \nabla f(x), \nabla^2 f(x)\nabla f(x) \rangle}{\|\nabla f(x)\|^2} \le \frac{\|\nabla f(x)\|\,\|\nabla^2 f(x)\nabla f(x)\|}{\|\nabla f(x)\|^2} \le \|\nabla^2 f(x)\|,$ where $\|\nabla^2 f(x)\|$ is the maximum Hessian eigenvalue (assumed bounded under standard smoothness).
> Assumption 5.11 is weaker because (a) it bounds only the directional second derivative and (b) it may hold with smaller constants; for example, Example 5.14 gives a real-world case with $L_0=0$. Since Assumption 5.11 is a relaxation, it is also automatically satisfied for some $(L_0,L_1)$ whenever standard smoothness holds.
> We could not recognize any issue in Line 284; could the reviewer please specify?

---

> ### Author Response · Authors · 2025-11-16
>
> >**Besides, some results (Corollary 5.7) in the paper contain strong assumptions on the trajectory, which I find unacceptable**
> Corollary 5.7 does not impose any assumptions on the trajectory. If the reviewer believes that a specific trajectory condition is “unacceptable,” or that a particular rate is missing, we would appreciate a pointer to the exact line or equation so that we can address it precisely. In Lines 186-189 we clearly state that the standard smoothness Assumption 5.5 holds due to the monotonicity of the methods and the assumptions made in Corollary 5.7. Beyond the assumptions made in Corollary 5.7, Assumption 5.11 relaxes the standard smoothness (as explained above). There are no additional assumptions in the deterministic part. Note that the vanishing directional curvature (10) is included primarily for illustration and is intentionally not stated as an assumption. Further, Assumption 6.1 concerns the stochastic case; immediately below it we cite unified analyses indicating that these assumptions are standard.
>
> >**some results (Theorem 6.3) are incomplete and do not give a convergence rate.**
> As discussed below Theorem 6.3, each resulting inequality recovers one of two regimes in the nonconvex setting: convergence of SGD or convergence of stochastic cubic Newton; the gradient norm vanishes as $1/k$ or as $1/k^{1.5}$, respectively. By extrapolating from the convex setting, we expect faster local convergence. In practice, our experiments show that the approach occasionally allows larger step sizes that dramatically improve convergence while preserving stability. However, combining these two bounds into a single rate appears challenging and, in our view, does not limit the contribution.
>
> >**The experiments in Section 7mseem cherry-picked. The results from the appendix suggest the proposed stepsize typically underperforms standard accelerated gradient descent.**
> The section is intentionally named **Selected experiments**, and we refer to the extended experiments in the first sentence. The comparison with accelerated methods is not appropriate, since they have provably faster rates. However, on severe benchmarks such as covtype, w8a, and ill-conditioned logsumexp, CaCuAdGD outperforms accelerated methods. We respectfully but strongly disagree that there is systematic underperformance. Moreover, CaCuAdGD is non-accelerated, so one can expect even faster convergence for an accelerated variant. Also, HVPs are proposed but can be replaced with vector-Hessian-vector products, which do not require backpropagation and can be computed even faster using forward passes only.
> We would like to point out that a fair comparison should first consider **non-accelerated adaptive methods**, such as AdGD or Polyak step sizes. These methods work very well on a variety of problems, adapt to local curvature, and do not incur computational overhead such as backtracking and HVP computations. Nevertheless, CaCuAdGD consistently outperforms these methods, which indicates a considerable contribution and validates the significance of the proposed technique.
>
> >**I don't feel I understand what "casual" in the title means. Does it come from the motivation on line 41**
> By “casual” we mean on-demand use of second-order information: the proposed methods avoid second-order work whenever possible and invoke it only when it is expected to help. Since it is difficult to predict a priori when this will be beneficial, the use of second-order computations is irregular (that is, occasional).
>
> >**Except for (AccCaCuN), the algorithms developed in the paper search along the gradient direction. Is there any intuition that the resulting algorithm can outperform vanilla gradient descent?**
> Although the search direction is the gradient, curvature information enters through the step size. This allows larger safe steps when curvature is mild and smaller steps when it is high, which can outperform vanilla gradient descent. The formalization is given in Section 3 (CRN-like descent result), and the intuition and comparison are discussed in Section 5.4 (Comparison with AdGD). We are happy to provide further details if helpful.
>
> >**Line 356 The assumption is not correct**
> Thanks for spotting a typo, it should be $x^k$ instead of $x$.
>
> >**Finally, I noticed a number of typos and notation inconsistencies throughout the paper. Overall, I don't think the paper can be published at ICLR in its current form.**
> We would appreciate a pointer to the exact line/equation so we can address it precisely. We are deeply grateful for spotting typos and other minor issues.
>
> [1] Gluon: Making Muon & Scion Great Again!(Bridging Theory and Practice of LMO-based Optimizers for LLMs) A Riabinin, E Shulgin, K Gruntkowska, P Richtárik

---

> ### Comment · Reviewer_LXQv · 2025-11-24
>
> Thank you for the response. Below, I detail my comments.
>
> 1. $O(1/K^2)$ rate
>
>    - **Section 5.1**. Although no theorem is stated in **Section 5.1**, the whole section 5.1 (especially the conclusion on line 267) relies on the condition (10), which is essentially an assumption. Otherwise, the $O(1/K^2)$ rate could be misleading to the readers. I would suggest you make these implicit assumptions clear. You might also consider formalizing (10) using assumptions similar to those presented in [1].
>    - By "assumptions on the trajectory", I refer to the statement like **Lemma 5.6** and **Corollary 5.7**, where the $O(1/K^2)$ rate only holds for points such that $f(x) - f(x^\star) \geq \big[\frac{3}{2}\frac{L D^2}{\sqrt{H D^3}}\big]^2$. It suggests the acceleration effect stops at some constant accuracy. Morever, this result is counterintuitive: imagine you are optimizing a simple convex quadratic function, then $H = 0$, and the conditions in **Lemma 5.6** and **Corollary 5.7** won't hold since the RHS of the condition $\frac{3}{2}\frac{L D^2}{\sqrt{H D^3}}$ goes to $+\infty$. I don't think this kind of convergence result is acceptable, since it suggests the algorithm performs worse on easier functions.
>
> 2. Smoothness terminology
>
>    To my knowledge, in the optimization literature [2], when one says $f$ is $L$-smooth, then it means $f$ has Lipschitz continuous gradient. Similarly, when you say $\nabla f$ is smooth, then $\nabla^2 f$ is Lipschitz continuous. So when you say Hessian smoothness, you are discussing continuity of $\nabla^3 f $. Here, precisely, I believe the algorithm is estimating the smoothness of $f$ along the gradient direction.
>
> 3. Incomplete convergence result
>
>    I believe it's better to write down the final convergence rate (complexity), instead of leaving a recursive relation to the readers as in **Theorem 6.3**. Please be precise.
>
> 4. Experiments
>
>    I appreciate your honesty about the experiments. But it does not address my concerns: a new algorithm should satisfy at least one of 1) practical efficiency; 2) strong theoretical performance, to be of interest. Given the current theoretical results and experiments, I still don't think they are satisfactory with respect to either criterion.
>
> I agree that this paper probably contains interesting insights that bring potential for future research. However, its current presentation is not there yet. I believe the paper still requires a major revision before meeting the publication standard of ICLR.
>
> **References.**
>
> [1] Fox, C., Mishkin, A., Vaswani, S., & Schmidt, M. (2025). Glocal Smoothness: Line Search can really help!. *arXiv preprint arXiv:2506.12648*.
>
> [2] Nesterov, Y. (2018). *Lectures on convex optimization* (Vol. 137). Berlin: Springer International Publishing.

---

> > ### Author Response · Authors · 2025-11-24
> >
> > Thank you for your response. We address your comments below.
> > >**Section 5.1. Although no theorem is stated in Section 5.1, the whole section 5.1 (especially the conclusion on line 267) relies on the condition (10), which is essentially an assumption. Otherwise the $O(1/K^2)$ rate could be misleading to the readers. I would suggest you make these implicit assumptions clear. You might also consider formalizing (10) using assumptions such as presented [1].**
> > >
> > >We are open to the suggestion and will clarify (10) as an illustrative assumption. This is a clarity tweak, not a technical flaw, and **cannot justify rejection**. Our main $O(1/k^2)$ result is under Assumption 5.11 with $L_0=0$ (Ex. 5.14; Cor. 5.15).
> >
> >
> > > **By "assumptions on the trajectory", I refer to the statement like Lemma 5.6 and Corollary 5.7, where there \(O(1/K^2)\) rate only holds for points such that \(f(x) - f(x^\star) \ge \tfrac{3}{2}\tfrac{L D^2}{\sqrt{H D^3}}\). It suggests the acceleration effect stops at some constant accuracy. Moreover, this result is counterintuitive: imagine you are optimizing a simple convex quadratic function, then \(H = 0\), and the conditions in Lemma 5.6 and Corollary 5.7 won't hold since the RHS of the condition \(\tfrac{3}{2}\tfrac{L D^2}{\sqrt{H D^3}}\) goes to \(+\infty\). I don't think this kind of convergence result is acceptable, since it suggests the algorithm performs worse on easier functions.**
> > >
> > >1. Lemma 5.6 / Cor. 5.7 / Theorem 5.8 do **not** impose “trajectory” assumptions; they **demarcate the regime** corresponding to the $O(1/k^2)$ phase. If the target accuracy $\varepsilon$ is large enough ($\varepsilon > \tfrac{3}{2}\tfrac{L D^2}{\sqrt{H D^3}}$, see Lines 316,331), then the converge as $O(1/k^2)$. Below that threshold the guarantee **reverts to $O(1/k)$** given by Theorem 5.8. This is not a flaw; it is **the result we present** -- the lemma partition accuracy targets: low final accuracy permit larger safe steps (yielding $O(1/k^2)$); finer targets recover the standard $O(1/k)$.
> > >
> > >2. On $H=0$ (quadratics). Here $H$ (Hessian smoothness constant) is zero, so the “accelerated” phase condition is non-existent. This does **not** mean worse behavior; the usual smooth convex analysis applies -- global $O(1/k)$ given by Theorem 5.8.
> > >
> > > **There is no mathematical inconsistency**; the statements and formulations are correct. We will add explicit clarifications to make the two-phase interpretation immediate. This is a presentation tweak, not a technical flaw, and **cannot justify rejection**.
> >
> >
> > > **To my knowledge, in the optimization literature [2], when one says \(f\) is smooth, then it means \(f\) has Lipschitz continuous gradient. Similarly, when you say \(\nabla f\) is smooth, then \(\nabla^2 f\) is Lipschitz continuous. So when you say Hessian smoothness, you are discussing continuity of \(\nabla^3 f\). Here, precisely, I believe the algorithm is estimating the smoothness of \(f\) along the gradient direction.**
> > >
> > >1. We use precise terms and avoid the **ambiguous** “smooth” as an assumption:
> > >
> > > - $f$ is **$L$-smooth** (defined at Line 305) = $f$ has **$L$-Lipschitz gradient**. This is a standard term appearing for example at page 2 in Zhang, Jingzhao, et al. "Why gradient clipping accelerates training: A theoretical justification for adaptivity." arXiv preprint arXiv:1905.11881 (2019).
> > >
> > > - **Hessian-Lipschitz** = **Lipschitz Hessian**
> > > - Similarly to $f$ $L$-smoothness, $\nabla f$ **$2H$-smoothness** refers to the standard Hessian-Lipschitz condition with constant $2H$ (Assumption 2.1).
> > >
> > >2. The algorithm estimates a Hessian-Lipschitz constant $2H$. It also adapts to **directional curvature along the gradient** via $\langle \nabla f(x), \nabla^2 f(x)\nabla f(x)\rangle / \|\nabla f(x)\|^2$.
> > >
> > >3. Some papers use $H$ as “the smoothness constant” (page 5 in Mishchenko, Konstantin. "Regularized Newton method with global convergence." SIAM Journal on Optimization 33.3 (2023): 1440-1462). In our paper, **“$2H$-smooth”** always means **Hessian-Lipschitz with constant $2H$** (Assumptions 2.1).
> > >
> > >**Edit (typos only).** We re-checked the paper and fixed several related typos (e.g., Line 288). However, $L$-smoothness and $2H$-smoothness” unambiguously refer to **Assumptions 2.1** and **5.5**, respectively, since the symbols $L$ and $2H$ are used consistently and uniquely throughout. This is an editorial cleanup, not a technical flaw, and **cannot justify rejection**.

---

> > ### Author Response · Authors · 2025-11-24
> >
> > > **I believe it's better to write down the final convergence rate (complexity), instead of leaving a recursive relation to the readers as in Theorem 6.3. Please be precise.**
> > >
> > >Our **claims are precise** and match Theorem 6.3. The theorem gives two per-iteration bounds that **each** yield an explicit rate **when that regime holds for all iterations**:
> > > - SGD-like regime ⇒ \(\|\nabla f(x^k)\|\) decreases as \(O(1/k)\).
> > > - cubic-Newton–like regime ⇒ \(\|\nabla f(x^k)\|\) decreases as \(O(1/k^{3/2})\).
> > >
> > >In **mixed mode** (the method may switch regimes across iterations), a single telescoped complexity bound is nontrivial (discussed below Theorem 6.3) and, to our knowledge, open; we intentionally do **not** over-claim.
> > >
> > > **Presentation edit.** We will add a short remark that states the two explicit rates when a single regime holds uniformly. This is a presentation edit, not a technical flaw, and **cannot justify rejection**.
> >
> >
> >
> >
> >
> > >**I appreciate your honesty about the experiments. But it does not address my concerns: a new algorithm should be satisfy at least one of 1) practical efficiency; 2) strong theoretical performance, to be of interest. Given the current theoretical results, I still don't think the current results are satisfactory in terms of either of these two criteria.**
> > >  - Criterion (1): **practical efficiency.** On several hard benchmarks (e.g., covtype, w8a, ill-conditioned logsumexp) our method matches or exceeds strong non-accelerated adaptives (AdGD/Polyak) and often closes the gap to accelerated methods; HVPs are cheap in modern frameworks and can be replaced by vHv (computable with only forward passes).
> > >
> > > - Criterion (2): **theory.** We recover the standard worst-case rate $O(1/k)$ for non-accelerated methods. We give a **new global $O(1/k^2)$ regime** under Assumption 5.11 with $L_0=0$ (Cor. 5.15) and the result in the nonconvex stochastic setting (Thm. 6.3). This is both **conceptual** (identify a realistic regime for $O(1/k^2)$ with first-order steps + occasional curvature) and **practical** (larger safe steps via HVP gating).
> > >
> > >We believe the paper meets both criteria. Empirically, CaCuAdGD consistently outperforms strong non-accelerated baselines and is competitive with accelerated methods on hard instances; theoretically, we provide the standard non-improvable $O(1/k)$ guarantees and a proved local/global $O(1/k^2)$ regime (global under Assumption 5.11 with $L_0=0$). The absence of an accelerated variant does not diminish these results. Accordingly, the **rejection rating does not reflect the paper’s contributions**; we respectfully **request a score reconsideration**.
> >
> >
> >
> > >**I agree that this paper probably contains interesting insights that bring potential for future research. However, its current presentation is not there yet. I believe the paper still requires a major revision before meeting the publication standard of ICLR.**
> > >We thank the reviews for pointing out the presentation issues. However, no technical flaws were identified. Moreover, it is explicitly noted that  the paper “probably contains interesting insights with potential for future research. We will apply the clarity edits. In view of this, **we respectfully request a score reconsideration**.

---

### Official Review · Reviewer_Esyd · 2025-11-01

**Soundness:** 3
**Presentation:** 3
**Contribution:** 2
**Rating:** 4
**Confidence:** 3

**Summary:**

The paper proposes a ‘casual cubic Newton’ framework, where the algorithm decides whether to use second-order updates or cheaper normalized gradient updates based on a Hessian-vector-product (HVP) certificate, which allows algorithms inspired by the framework (CaCuN and AccCaCun) to enjoy similar convergence rates with cubic-regularized Newton, yet without forming or factorizing Hessians as often. Moreover, under a specific assumption, it is possible to construct an entirely HVP-based algorithm (CaCuAdGD), where the algorithm adapts to the local geometry via the HVP in the ‘certificate inequality’ and the backtracking of the constant $H$.

**Strengths:**

- The writing is clear and easy to read overall.
- The paper includes numerical experiments that support the theoretical results.

**Weaknesses:**

See **Questions.**

**Questions:**

- For CaCuN (and AccCaCuN) in Section 4, is it possible to quantify how much CRN steps we might expect to need? I think this should be a bit clearer that the algorithm will stay at the regime (with constants simplified to $c'$),
$$
f \left( y^k - \frac{\nabla f(y^k)}{\sqrt{c' \cdot H \| \nabla f(y^k) \|}} \right) \le f(y^k) - \frac{1}{c' \cdot \sqrt{H}} \| \nabla f(y^k) \|^{3/2}
$$
at least quite often to conclude that using CRN steps only outside this regime will really be computationally beneficial than vanilla methods and/or comparable with other efficient second-order methods like lazy Hessians (Doikov et al., 2023). Is it possible for the authors to quantify the Hessian (and gradient) oracle costs, at least on a high-level? Or could there be some specific cases where we can clearly expect fewer CRN steps?
    - In particular, when taking iterations other than CRN steps (i.e., excluding the Hessian computation steps), my intuition is that methods like lazy Hessians, which both preserves the Hessian structure itself but uses matrix-vector products for most steps, might be better than using a first-order method based on a upper-bound-ish scalar $H$, which is typically a bit more conservative. I didn’t check this in detail, could the authors elaborate on this?
    - I might be wrong, but isn’t this ‘HVP-based certificate’ is actually fully first-order computable if we take $H$ as constants, not matrices (which could actually be slightly better than Hessian-vector products)? This is a minor thing, but the term is a bit confusing to me as I can only see HVPs in Algorithm 3 (CaCuAdGD).
- For CaCuAdGD in Section 5, while it is nice that we can enjoy convergence only with first-order oracles and Hessian-vector products, it feels like that the assumption in $(10)$ is essentially just tailored to make the above regime to be true. As there are a few examples (cubic and logistic functions) mentioned that fall into this category, is there a better reason to consider this assumption other than merely a necessary condition for similar ideas as in the previous section to work? (This could also possibly be related to the first question, as we might be able to assert that CaCuN type algorithms will require very few Hessian oracles for the cubic and logistic functions.)

Doikov et al., 2023. Second-order optimization with lazy Hessians.

---

> ### Author Response · Authors · 2025-11-16
>
> We thank the reviewer for valuable comments. We find them intuitive and motivating for further studies and discussions. We are pleased to address the questions and kindly ask the reviewer to reconsider the score.
>
> >**For CaCuN (and AccCaCuN) in Section 4, is it possible to quantify how much CRN steps we might expect to need? I think this should be a bit clearer that the algorithm will stay at the regime (with constants simplified to $c'$), $$ f \left( y^k - \frac{\nabla f(y^k)}{\sqrt{c' \cdot H | \nabla f(y^k) |}} \right) \le f(y^k) - \frac{1}{c' \cdot \sqrt{H}} | \nabla f(y^k) |^{3/2} $$ at least quite often to conclude that using CRN steps only outside this regime will really be computationally beneficial than vanilla methods and/or comparable with other efficient second-order methods like lazy Hessians (Doikov et al., 2023). Is it possible for the authors to quantify the Hessian (and gradient) oracle costs, at least on a high-level? Or could there be some specific cases where we can clearly expect fewer CRN steps?**
> We do not currently have a theoretical result on “expected” oracle costs. We can, however, comment on experimental behavior. CaCuN-type methods are effective in non-adaptive settings, i.e., when a global $H$ is known. In realistic scenarios with adaptive methods available (see Appendix C), our methods typically do not require second-order computations in the first iterations. Later, we observe that the local value $H_k$ can change substantially: after a successful step (large objective decrease), $H_k$ often increases sharply, after which a few gradient steps are taken until $H_k$ recovers to the local value. The algorithm checks the first-order certificate at every iteration; in our runs this adds little overhead and causes no significant gain in the overall performance. We plan to add large-scale experiments which may reduce the relative amount of work for the checks. Finally, unlike Doikov et al. (2023), we preserve CRN rates; we expect that a provable quantification may be possible, potentially at the cost of an additional factor similar to $\sqrt{m}$ in their analysis.
>
>
> >**In particular, when taking iterations other than CRN steps (i.e., excluding the Hessian computation steps), my intuition is that methods like lazy Hessians, which both preserves the Hessian structure itself but uses matrix-vector products for most steps, might be better than using a first-order method based on a upper-bound-ish scalar $H$, which is typically a bit more conservative. ...could the authors elaborate on this?**
> Regarding lazy Hessians, our experience is that an accurate Hessian is not needed to recover after a sharp increase in $H_k$. Adaptation to the local value $H_k$ is the key driver of practical performance. The adaptive approach in Doikov et al. (2023) scales the local constant as $m H_k$; for larger $m$, Hessian regularization may become conservative, which aligns with our motivation. Combining lazy Hessians with CaCuTe may be a promising direction.
>
>
>
>
> >**I might be wrong, but isn’t this ‘HVP-based certificate’ is actually fully first-order computable if we take $H$ as constants, not matrices (which could actually be slightly better than Hessian-vector products)? This is a minor thing, but the term is a bit confusing to me as I can only see HVPs in Algorithm 3 (CaCuAdGD).**
> If the point is that an HVP can have similar cost to a gradient, we agree and note this at Line 107. We avoid calling it “first order,” since the Hessian is involved. Moreover, CaCuTe requires only vector–Hessian–vector products; these scalar quantities can be computed even more cheaply, using forward passes only.
>
> >**For CaCuAdGD in Section 5, while it is nice that we can enjoy convergence only with first-order oracles and Hessian-vector products, it feels like that the assumption in (10) is essentially just tailored to make the above regime to be true. As there are a few examples (cubic and logistic functions) mentioned that fall into this category, is there a better reason to consider this assumption other than merely a necessary condition for similar ideas as in the previous section to work? (This could also possibly be related to the first question, as we might be able to assert that CaCuN type algorithms will require very few Hessian oracles for the cubic and logistic functions.)**
> Equation (10) is intentionally not stated as an assumption; it is non-standard and serves primarily for illustration, as it allows us to obtain a global $O(k^{-2})$ rate with a short argument. The substantive reason is the relaxation of standard smoothness via Assumption 5.11, which covers the usual $L$-smoothness and yields a global $O(k^{-2})$ rate when $L_0=0$. This assumption is realistic; for example, it holds for separable logistic regression loss (Example 5.14). The idea of introducing alternative assumptions to enable “expected” oracle-cost quantification is valuable; we do not have a result at present and plan to study this direction.

---

### Author Response · Authors · 2025-12-02
**Note to Area Chair regarding Review soLp**

The review’s central claims (“unclear contribution,” “insufficient rigor”) are **not supported by any concrete evidence**:

- The paper’s contributions are clearly itemized in Section 1.3.
- The review does not identify any incorrect theorem, false assumption, or invalid derivation.
- All results are formally stated (Thm. 4.1, Thm. 5.8, Cor. 5.15, Thm. 6.3) under explicit assumptions (Secs. 2, 5, 6), with complete proofs in the appendices. The review does not engage with any specific line of these proofs or indicate where the arguments would fail.

Instead, the review largely consists of high-level statements and clarifying questions about basic concepts, which point to issues of understanding and exposition rather than to any lack of rigor in the results. In the absence of specific technical faults, we respectfully submit that a **reject recommendation based on unsubstantiated criticism** should carry limited weight.

The remaining concerns focus on **clarity/presentation**, not correctness. We **disagreed** that the results were unverifiable as written, but we have rewritten the relevant parts (changes marked in teal in the revised version) so that every step is even easier to follow:

- **Contribution cross-referencing.** Each stated contribution now points to the section where it is developed and cross-references the appendix where its proof is located.
- **Standard assumptions.** Concepts like $L$-smoothness, Hessian–Lipschitz continuity, convexity, and bounded level sets are standard and were **already** cited; we now place the citations **immediately under the assumptions**.
- **Certificate mechanism.** The presentation is simplified: we either explicitly check the descent inequality or use the step-size rule that implies it.
- **CaCuN full proof.** We added a **standalone CaCuN theorem** with a complete, self-contained proof. **As initially explained**, the proof follows Nesterov’s approach; we only add a simple one-line implication from the descent inequality used for the gradient branch.
- **Minor but explicit clarifications.** We include standalone lemmas showing how the descent inequalities yield the stated rates (previously only referenced). We also clarify that Eq. (10) is **illustrative**, and the formal contribution is **Assumption 5.11**.
- **Typos.** All noted typos are fixed.

**None of these changes alter any claim.** These edits are **purely expository**; statements, assumptions, and rates are unchanged. We ask that the decision focus on the **technical contributions**, not presentation preferences.

---

### Meta-Review · Area_Chair_2Y6o · 2026-01-16

**Summary:**

This paper solves an unconstrained optimization problem with a twice-differentiable function $f$ with Lipschitz Hessian. The goal is to obtain an accelerated $O(k^{-2})$ rate while avoiding Hessian evaluation and inversion. The authors accomplish this either under condition in Eq (10) or $L_0, L_1$ Lipschitzness with $L_0=0$. All the reviewers and the authors agree that Eq. (10) is too restrictive. On the other hand, the latter assumption is also rather restrictive and it's not clear what the improvement is with nonzero $L_0$. Moreover, all the reviewers and myself agree that the exposition needs to be improved. Unfortunately, this is significant for a conference paper because there is no way of checking the final version (unlike a journal submission). As a result, a major revision required for exposition is a valid reason for rejection. Given that there are concerns about the contribution (strength of assumptions for the accelerated rate) and the exposition, I recommend rejection.

**Reviewer Concerns:**

Some of the reviewers concerns about the exposition are resolved. However, an important concern shared by Reviewer Esyd, Reviewer LXQv and Reviewer soLP is that the accelerated rate requires Eq (10) or some Lipschitzness assumptions which are rather strong. The authors made a good case to highlight their Lipschitz assumption holds for some applications, but they are missing a comparison between their results and their assumption as well as similar results in the literature with their corresponding assumptions. A clear positioning and a clearer exposition are not provided during the review period, hence these important concerns are not addressed completely.

**Reviewer Scores:**

Both Reviewer LXQv and Reviewer soLP engaged in discussion and their concerns were not resolved. Looking at their concerns and the responses, I think that their scores would not have changed even with a longer discussion period.

Reviewer w1tX suggested acceptance with a score of 6 and I believe they wouldn't have changed their score even with a longer discussion period.

Reviewer Esyd suggested rejection with a score of 4. They had shared similar concerns to Reviewer LXQv and Reviewer soLP but were more positive. Under a longer discussion period, their score may have increased to a 6, but given that I think the concerns raised by this reviewer as well as Reviewer LXQv and Reviewer soLP are significant, I recommended rejection.

---

### Decision · Program_Chairs · 2026-01-26

Reject